# Democratizing Fine-grained Visual Recognition with Large Language Models

**Mingxuan Liu[1], Subhankar Roy[4], Wenjing Li[3,6*], Zhun Zhong[3,5*], Nicu Sebe[1], Elisa Ricci[1,2]**

[1] University of Trento, Trento, Italy     [2] Fondazione Bruno Kessler, Trento, Italy
[3] Hefei University of Technology, Hefei, China     [4] University of Aberdeen, Aberdeen, UK
[5] University of Nottingham, Nottingham, UK     [6] University of Leeds, Leeds, UK

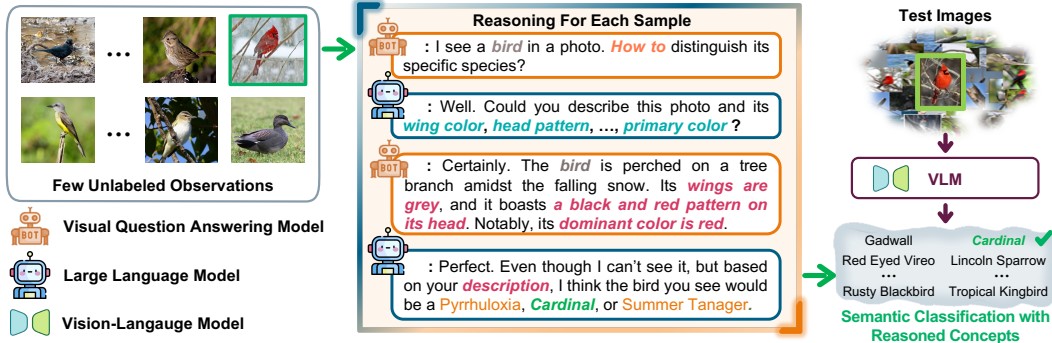

Figure 1: An overview of our proposed fine-grained visual recognition (FGVR) pipeline. **Left**: Given few unlabelled images we exploit visual question answering (VQA) and large language models (LLM) to reason about subordinate-level category names without requiring expert knowledge. **Right**: At inference, we utilize the reasoned concepts to carry out FGVR via zero-shot semantic classification with a vision-language model (VLM).

## Abstract

Identifying subordinate-level categories from images is a longstanding task in computer vision and is referred to as fine-grained visual recognition (FGVR). It has tremendous significance in real-world applications since an average layperson does not excel at differentiating species of birds or mushrooms due to subtle differences among the species. A major bottleneck in developing FGVR systems is caused by the need of high-quality paired expert annotations. To circumvent the need of expert knowledge we propose **F**ine-grai**n**ed Sema**n**tic Cat**e**gory **R**easoning (FineR) that internally leverages the world knowledge of large language models (LLMs) as a proxy in order to reason about fine-grained category names. In detail, to bridge the modality gap between images and LLM, we extract part-level visual attributes from images as text and feed that information to a LLM. Based on the visual attributes and its internal world knowledge the LLM reasons about the subordinate-level category names. Our training-free FineR outperforms several state-of-the-art FGVR and language and vision assistant models and shows promise in working in the wild and in new domains where gathering expert annotation is arduous.

## 1 Introduction

Fine-grained visual recognition (FGVR) is an important task in computer vision that deals with identifying subordinate-level categories, such as species of plants or animals (Wei et al., 2021). It is challenging due to the fact that different species of birds can differ in subtle attributes, such as the Lincoln's Sparrow mainly differs from Baird's Sparrow in the coloration of the breast pattern (see Fig. 2). Due to small inter-class and large inner-class variations, the FGVR methods typically require auxiliary information namely part annotations (Zhang et al., 2014), attributes (Vedaldi et al., 2014), natural language descriptions (He & Peng, 2017) collected with the help of experts in the respective fields. Hence, this expensive expert annotation presents as a bottleneck and prevents FGVR in new domains from being rolled out as software or services to be used by common users.

On the contrary encyclopedia of textual information about plants and animals can be found on the internet that document at great lengths about the characteristics and appearance of each species. In

---

*Corresponding authors.     Code is available at `https://projfiner.github.io`

other words, modern day Large Language Models (LLMs) (*e.g.*, ChatGPT (OpenAI, 2022)), trained on enormous internet-scale corpuses, already encode in its weights the expert knowledge that is quintessential in FGVR. In this work we ask ourselves, *can we build FGVR systems by exploiting the unpaired auxiliary information already contained in LLMs*? The implications are tremendous as it endows an average layperson with the expert knowledge of a mycologist, ornithologist and so on.

Leveraging unlabelled FGVR dataset and the LLM to build a FGVR model is not straightforward as there exists a modality gap between the two. In detail, a LLM cannot *see*, as it can only ingest text (or prompts) at input and give text at output. To bridge this modality gap, we first use a general purpose visual question answering (VQA) model (*e.g.*, BLIP-2 (Li et al., 2023)) to extract a set of visual attributes from the images. Specifically, for every image in a bird dataset we prompt BLIP-2 to extract part-level descriptions of wing color, head pattern, tail pattern and so on. We then feed these attributes and their descriptions to a LLM, and prompt it to reason on a set of candidate class names using its internal knowledge of the world. As a final step, we

| | | |
|---|---|---|
| **Super-category:** | Bird | Bird |
| **Ground-truth:** | Lincoln's Sparrow | Baird's Sparrow |
| **BLIP-2:** | Bird | Bird |
| | Sparrow | Sparrow |
| **LLaVA:** | Bird | Bird |
| | Sparrow | Sparrow |
| **LENS:** | Bird | Bird |
| | Vesper Sparrow | Clay-colored Sparrow |
| **MiniGPT-4:** | Bird | Bird |
| | White-throated Swainson Hawk | White-rumped Sandpiper |
| **FineR (Ours):** | Bird | Bird |
| | Lincoln's Sparrow | Baird's Sparrow |

Figure 2: Comparing our proposed FineR with the state-of-the-art visual question answering models: BLIP-2 (Li et al., 2023), LLaVA (Liu et al., 2023a), LENS (Berrios et al., 2023), and MiniGPT-4 (Zhu et al., 2023b).

use the candidate class names to construct a semantic classifier, as commonly used in the vision-language models (VLMs) (*e.g.*, CLIP (Radford et al., 2021)), to classify the test images. We call our method **F**ine-gra**i**ned Sema**n**tic Cat**e**gory **R**easoning (FineR) as it discovers concepts in unlabelled fine-grained datasets via reasoning with LLMs (see Fig. 1).

Through our proposed FineR system we bring together the advances in VQA, LLMs and VLMs to address FGVR that was previously not possible without accessing high-quality paired expert annotations. Since our proposed FineR does not require an expert's intervention, it can be viewed as a step towards *democratizing* FGVR systems by making them available to the masses. In addition, unlike many end-to-end systems our FineR is interpretable due to its modularity. We compare our method with FGVR, vocabulary-free classification and multi-modal language and vision assistant models (*e.g.*, BLIP-2) and show its effectiveness in the challenging FGVR task. As shown in Fig. 2 our FineR is more effective than the state-of-the-art methods, which are accurate at predicting the super-category but fail at identifying the subordinate categories. Unique to our FineR, it works in a few-shot setting by accessing only a few unlabelled images per category. This is a very pragmatic setting as fine-grained classes in real-world typically follow an imbalanced class distribution. Finally, our method is completely *training-free*, which is the first of its kind in FGVR. Extensive experiments on multiple FGVR benchmarks show that our FineR is far more effective than the existing methods.

In summary our contributions are: (**i**) We introduce the challenging task of recognizing fine-grained classes using only a *few* samples per category, where *neither* labels *nor* paired expert supervision is available. (**ii**) We propose the FineR system for FGVR that exploits a cascade of foundation models to reason about possible subordinate category names in an image. (**iii**) With extensive experiments on multiple FGVR benchmarks we demonstrate that our FineR is far more effective than the existing approaches. We additionally introduce a new fine-grained benchmark of Pokemon characters and show that FineR exhibits the potential to work truly in the-the-wild, even for *fictional* categories.

## 2 METHODOLOGY

**Problem Formulation.** In this work, we study the problem of fine-grained visual recognition (FGVR) given an unlabeled dataset $\mathcal{D}^{\text{train}}$ of fine-grained objects. Different from the traditional FGVR task (Wei et al., 2021) we do not use any label or paired auxiliary information annotated by an expert (Choudhury et al., 2023), and $\mathcal{D}^{\text{train}}$ contains a few unlabelled samples per category (Li et al., 2022). Our goal is to learn a model that can assign correct labels to the images in the test set $\mathcal{D}^{\text{test}}$. In detail, when presented with a test image we output a semantic class name $c \in \mathcal{C}$, where $\mathcal{C}$ is the set of class names that are not known a priori, making it vocabulary-free (Conti et al., 2023). Our proposed FineR first aims to generate the set $\mathcal{C}$ and then assigns the test image to one of the classes. As an

example, opposed to the FGVR methods that output a class-index at inference, our FineR directly outputs a semantic class name (*e.g.*, Blue-winged Warbler). Next we discuss some preliminaries about the pre-trained models that constitute our proposed FineR.

## 2.1 PRELIMINARIES

**Large Language Models (LLMs)**, such as ChatGPT (OpenAI, 2022) and LLaMA (Touvron et al., 2023), are auto-regressive models that are trained on large internet-scale text corpuses possess a wealth of world knowledge encoded in their weights. These models can be steered towards a particular task by conditioning on demonstrations of input-output example pairs, relevant to the task at hand, without the need of fine-tuning the weights. This emergent behaviour is referred to as in-context learning (ICL) (Dong et al., 2022; Brown et al., 2020). The LLMs have also demonstrated adequate reasoning abilities in answering questions when prompted with well-structured prompts (Wei et al., 2022), another emergent behaviour that we exploit in this work. Formally, a LLM $h^{\text{llm}} : \rho^{\text{llm}} \mapsto T^{\text{llm}}$ takes a text sequence (or LLM-prompt) $\rho^{\text{llm}}$ at input and maps it to a text output sequence $T^{\text{llm}}$.

**Vision-Language Models (VLMs)**, such as CLIP (Radford et al., 2021) and ALIGN (Jia et al., 2021), consist of image and text encoders that are jointly trained to predict the correct pairings of noisy image and text pairs. Due to well-aligned image-text representation space, they demonstrate excellent zero-shot transfer performance on unseen datasets, provided the class names of the test set is known a priori (Radford et al., 2021). A VLM $h^{\text{vlm}}$ maps an image $\mathbf{x}$ to a category $c$ in a given vocabulary set $\mathcal{C}$ based on the maximum cosine similarity:

$$\hat{c} = \arg \max_{c \in \mathcal{C}} \langle \mathcal{E}_{\text{img}}(\mathbf{x}), \mathcal{E}_{\text{txt}}(c) \rangle, \tag{1}$$

where $\hat{c}$ is the predicted class, and $\mathcal{E}_{\text{img}}$ and $\mathcal{E}_{\text{txt}}$ denote the vision and text encoder of the VLM, respectively. $\langle \cdot, \cdot \rangle$ represents the cosine similarity function.

**Visual Question Answering (VQA)** models, such as BLIP-2 (Li et al., 2023) and LLaVA (Liu et al., 2023a), that combine LLM and VLM in a single framework, learn to not only align the visual and text modalities, but also to generate the caption of the image-text pair. This visual-language pre-training (VLP) is then used to perform VQA as one of the downstream tasks. In detail, at inference a VQA model $h^{\text{vqa}} : (I, \rho^{\text{vqa}}) \mapsto T^{\text{vqa}}$ receives an image $I$ and a textual question (or VQA-prompt) $\rho^{\text{vqa}}$ as inputs, and outputs a textual answer $T^{\text{vqa}}$ that is conditioned on both the visual and textual inputs.

While these VQA models excel at identifying super-categories and general visual attributes (*e.g.*, color, shape, pattern) of objects when prompted with a question, they fail to recognize many subordinate-level categories (see Fig. 2) due to lack of well-curated training data. We show how the aforementioned pre-trained models can be synergistically combined to yield a powerful FGVR system.

## 2.2 FINER: FINE-GRAINED SEMANTIC CATEGORY REASONING SYSTEM

To recall, our goal is to recognize the fine-grained categories using unlabelled images in $\mathcal{D}^{\text{train}}$, without having access to expert annotations and the ground-truth categories in $\mathcal{C}$. Thus, the challenges in such a FGVR task are to first identify (or discover) the classes in $\mathcal{C}$ and then assign a class name $c \in \mathcal{C}$ to each instance in $\mathcal{D}^{\text{test}}$. The idea underpinning our proposed **F**ine-grained Sema**n**tic Cat**e**gory **R**easoning (FineR) is that the LLMs, which already encode the world knowledge about different species of animals and plants, can be leveraged to reason about candidate class names $\widehat{\mathcal{C}}^*$ when prompted with visual attributes extracted from the VQA models. Concretely, FineR takes a few unlabeled images in $\mathcal{D}^{\text{train}}$ as input, and outputs the discovered class names $\widehat{\mathcal{C}}^*$. Subsequently, the discovered candidate class names and images in $\mathcal{D}^{\text{train}}$ are utilized to yield a multi-modal classifier $\mathbf{W}_{\text{mm}}$ to classify a test instance using a VLM. Therefore, our FineR aims to learn a mapping:

$$\mathcal{H}^{\text{FineR}} : \mathcal{D}^{\text{train}} \mapsto (\widehat{\mathcal{C}}^*, \mathbf{W}_{\text{mm}}),$$

where $\mathcal{H}^{\text{FineR}}$ is a meta-controller that encapsulates our system. FineR operates in three phases: (i) Translating Useful Visual Information from Visual to Textual Modality; (ii) Fine-grained Semantic Category Reasoning in Language; and (iii) Multi-modal Classifier Construction. An overview of our system is depicted in Fig. 3 and the details follow next.

### 2.2.1 TRANSLATING USEFUL VISUAL INFORMATION FROM VISUAL TO TEXTUAL MODALITY

Before we can tap into the expert world knowledge inherent to the LLMs for facilitating FGVR, we need a mechanism to translate the visual information from the images into an input format which the

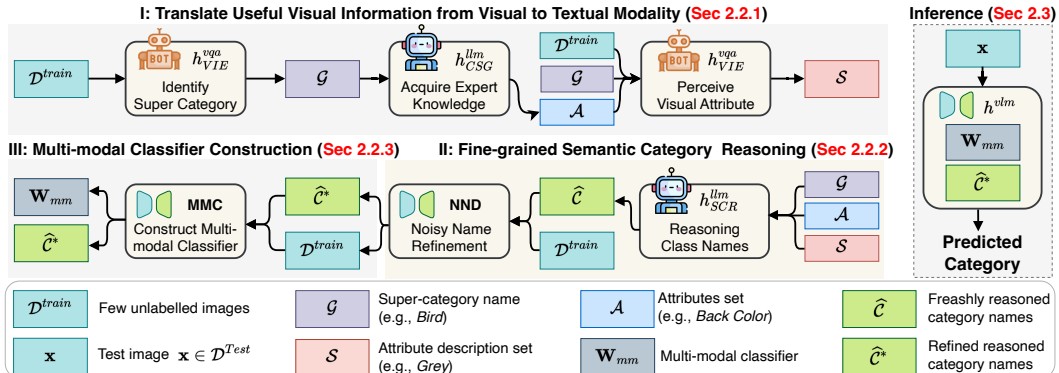

Figure 3: The pipeline of the proposed **F**ine-gra**i**ned Sema**n**tic Cat**e**gory **R**easoning (FineR) system.

LLM understands. Furthermore, we require an understanding about which visual cues are informative at distinguishing two fine-grained categories. For instance, what separates two species of birds (Lincoln's Sparrow vs Baird's Sparrow) is drastically different than what separates two types of mushrooms (Chanterelle vs Porcini). Motivated by this we design the following steps (see Fig. 3 top).

**Identify super-category.** To make our framework general purpose and free of any human intervention we first identify the super-category of the dataset $\mathcal{D}^{\text{train}}$. In detail, we use a VQA model as a Visual Information Extractor (VIE) $h_{\text{VIE}}^{\text{vqa}}$ for this purpose. The VIE is capable of identifying the super-category (e.g., "Bird") of the incoming dataset as it has been pre-trained on large amounts of data from super-categories. Formally, to get super-category $g_n$ we query the VIE for an image $\mathbf{x}_n \in \mathcal{D}^{\text{train}}$:

$$g_n = h_{\text{VIE}}^{\text{vqa}}(\mathbf{x}_n, \rho_{\text{identify}}^{\text{vqa}}),\tag{2}$$

where $\rho_{\text{identify}}^{\text{vqa}}$ is a simple **Identify VQA-prompt** of the format shown in Fig. 10, and $\mathcal{G} = \{g_1, g_2, \ldots, g_N\}$ is a set containing the super-category names corresponding to $N$ images in $\mathcal{D}^{\text{train}}$. To guide the VIE to output only super-category names we provide simple in-context examples as a list of names ("Bird", "Pet", "Flower", ...). Note that the ground-truth super-category name need not be present in the list (see App. B.2 for details about in-context examples).

**Acquire expert knowledge.** After having a general understanding of the dataset this step consists in identifying the useful set of attributes that set apart the subordinate-level categories. For instance, the "eye-color" attribute can be used to easily distinguish the bird species Dark-eyed Junco from the species Yellow-eyed Junco. To help the VIE in focusing on such targetted visual cues we tap into the expert knowledge of the LLMs, which is otherwise only restricted to experts. In other words, we essentially ask the LLM: given a super-category *what* to look for while distinguishing subordinate-level categories. In detail, we base our Acquire Expert Knowledge (AEK) module $h_{\text{AEK}}^{\text{llm}}$ on a LLM that takes an input a super-category $g_n$ and outputs a list of $M$ useful attributes:

$$\boldsymbol{a}_n = h_{\text{AEK}}^{\text{llm}}(g_n, \rho_{\text{how-to}}^{\text{llm}}),\tag{3}$$

where $\boldsymbol{a}_n = \{a_{n,1}, a_{n,2}, \ldots, a_{n,M}\}$ denote the $M$ attributes corresponding to the sample with index $n$, and **How-to LLM-prompt** $\rho_{\text{how-to}}^{\text{llm}}$ is the prompt, as shown in Fig. 10. The intuition behind the How-to LLM-prompt is to emulate *how to be an expert* and forms the crux of our FineR. All the attributes in the dataset are denoted as $\mathcal{A} = \{\boldsymbol{a}_1, \ldots, \boldsymbol{a}_N\}$.

**Perceive visual attributes.** With the attributes $\mathcal{A}$ available we leverage the VIE to extract the description for each attribute $a_{n,m}$ conditioned on the visual feature. For instance, if an attribute $a_{n,m}$ = "eye-color", then given the image of an unknown bird the VIE is asked to describe it's eye color, which is a much easier task compared to directly predicting the subordinate-level category. In addition, we hard-code a *general* attribute $a_0 =$ "General description of the image" and its prompt "Questions: Describe this image in details. Answer:" to extract the global visual information. Therefore, $\boldsymbol{a}_n \leftarrow \boldsymbol{a}_n \cup a_0$ and $M \leftarrow M + 1$. We found this is helpful to capture the useful visual information from the background like the habitat of a bird. Concretely, given an image $\mathbf{x}_n$, its super-category $g_n$ and the attribute $a_{n,m}$, the attribute description is given as:

$$s_{n,m} = h_{\text{VIE}}^{\text{vqa}}(\mathbf{x}_n, g_n, a_{n,m}, \rho_{\text{describe}}^{\text{vqa}}),\tag{4}$$

where $\rho_{\text{describe}}^{\text{vqa}}$ denotes the **Describe VQA-Prompt**, as illustrated in Fig. 10, and $\mathcal{S}_n = \{s_{n,1}, s_{n,2}, \ldots, s_{n,M}\}$ represents a set of $M$ attribute descriptions extracted from the image $\mathbf{x}_n$.

### 2.2.2 FINE-GRAINED SEMANTIC CATEGORY REASONING

At the end of the first phase (Sec. 2.2.1) we have successfully translated useful visual information into text, a modality that is understood by the LLM. In the second phase our goal is leverage a LLM and utilize all the aggregated textual information to reason about class names in the dataset and predict a candidate set $\widehat{\mathcal{C}}^*$, as illustrated in the middle part of Fig. 3.

**Reasoning class names.** To discover the candidate class names present in the dataset we build a Semantic Category Reasoner (SCR) module that contains a LLM under the hood. The goal of SCR to evoke reasoning ability of the LLM by presenting a well-structured prompt constructed with the attributes and descriptions obtained in Sec. 2.2.1. Formally, the SCR is defined by a function $h_{\text{SCR}}^{\text{llm}}$ that outputs a preliminary set of candidate classes $\widehat{\mathcal{C}}$ as:

$$\widehat{\mathcal{C}} = h_{\text{SCR}}^{\text{llm}}(\mathcal{G}, \mathcal{A}, \mathcal{S}, \rho_{\text{reason}}^{\text{llm}}), \tag{5}$$

where $\rho_{\text{reason}}^{\text{llm}}$ is the **Reason LLM-prompt** as shown in Fig. 10. The $\rho_{\text{reason}}^{\text{llm}}$ can be decomposed into two parts: (i) **Structured Task Instruction** is designed keeping in mind the FGVR task and is mainly responsible for steering the LLM to output multiple possible class names given an image, thereby increasing diversity in names. We find that when only one name is asked, LLMs tend to output general categories names (*e.g.*, "Gull") from a coarser but more common granularity. (ii) **Output Instruction** embeds the actual attribute description pairs extracted from the image, and task execution template to invoke the output (see App. B.2 for details).

**Noisy name refinement.** In the pursuit of increasing the diversity of candidate class names we accumulate in the set $\widehat{\mathcal{C}}$ a lot of *noisy* class names. It is mainly caused by the unconstrained knowledge in LLMs that is not relevant to the finite concepts in the $\mathcal{D}^{\text{train}}$ at hand. To remedy this, in addition to name deduplication, we propose a Noisy Name Denoiser (NND) module that uses a VLM (*e.g.*, CLIP). Specifically, NND uses the text encoder $\mathcal{E}_{\text{txt}}$ and the vision encoder $\mathcal{E}_{\text{img}}$ of the VLM to encode $\widehat{\mathcal{C}}$ and $\mathcal{D}^{\text{train}}$, respectively. Each image in $\mathcal{D}^{\text{train}}$ is assigned to a category $\hat{c} \in \widehat{\mathcal{C}}$ based on the maximum cosine similarity as in Eq. 1. The class names that are not selected by Eq. 1 constitute the noisy class names set $\mathcal{V} \subset \widehat{\mathcal{C}}$ and are eliminated. The refined candidate set is given as $\widehat{\mathcal{C}}^* = \widehat{\mathcal{C}} \setminus \mathcal{V}$.

### 2.2.3 MULTI-MODAL CLASSIFIER CONSTRUCTION

Given the refined class names in $\widehat{\mathcal{C}}^*$ we construct a text-based classifier for each class $c \in \widehat{\mathcal{C}}^*$ using the text-encoder $\mathcal{E}_{\text{txt}}$ of the VLM as:

$$\boldsymbol{w}_{\text{txt}}^c = \frac{\mathcal{E}_{\text{txt}}(c)}{||\mathcal{E}_{\text{txt}}(c)||_2}. \tag{6}$$

Since there can be ambiguity in some of the class names (Kaul et al., 2023) we also consider a vision-based classifier by using the images in $\mathcal{D}^{\text{train}}$. In detail, we first pseudo-label the images in $\mathcal{D}^{\text{train}}$ with $\widehat{\mathcal{C}}^*$ using the maximum cosine similarity metric, as outlined in Sec. 2.2.2. This results in $U_c$ pseudo-labelled images per class $c \in \widehat{\mathcal{C}}^*$. However, as $\mathcal{D}^{\text{train}}$ contains only a few images per class it might make the visual features strongly biased towards those samples. For example, only a few colors or perspectives of a car of make and model Toyota Supra Roadster is captured in those images. Thus, we apply a random data augmentation pipeline $K$ times on each sample $\mathbf{x}$ (see App. B.1 for details) and then compute the vision-based classifier for each class $c \in \widehat{\mathcal{C}}^*$ as:

$$\mathbf{w}_{\text{img}}^c = \frac{1}{U_c(K+1)} \left( \sum_{i=1}^{U_c} \frac{\mathcal{E}_{\text{img}}(\mathbf{x}_i^c)}{||\mathcal{E}_{\text{img}}(\mathbf{x}_i^c)||_2} + \sum_{j=1}^{K \times U_c} \frac{\mathcal{E}_{\text{img}}(\tilde{\mathbf{x}}_j^c)}{||\mathcal{E}_{\text{img}}(\tilde{\mathbf{x}}_j^c)||_2} \right), \tag{7}$$

where $\tilde{\mathbf{x}}$ is the augmented version of $\mathbf{x}$. We set $K = 10$ in all our experiments (see App. G.2 for an analysis on the sensitivity of $K$).

Being equipped with classifiers of two modalities, we construct a multi-modal classifier (MMC) for class $c$ that potentially can capture complementary information from the two modalities:

$$\mathbf{w}_{\text{mm}}^c = \alpha \mathbf{w}_{\text{txt}}^c + (1 - \alpha) \mathbf{w}_{\text{img}}^c, \tag{8}$$

where $\alpha$ is the hyperparameter that controls the degree of multi-modal fusion. We empirically set $\alpha = 0.7$ in all of our experiments (see App. G.1 for an analysis on the sensitivity of $\alpha$). Finally, the MMC for all the classes in $\widehat{\mathcal{C}}^*$ is given as $\mathbf{W}_{\text{mm}}$.

## 2.3 INFERENCE

As shown in Fig. 3 right, we replace the text-based classifier in VLM with the MMC $\mathbf{W}_{\text{mm}}$, and classify the test images $\mathbf{x} \in \mathcal{D}^{\text{test}}$ by $\hat{c} = \arg\max_{c \in \widehat{\mathcal{C}}^*} \langle \mathcal{E}_{\text{img}}(\mathbf{x}), \mathbf{w}_{\text{mm}}^c \rangle$, where $\hat{c}$ is the predicted class name having the highest cosine similarity score.

## 3 EXPERIMENTS

**Datasets.** We have conducted experiments on several fine-grained datasets that include Caltech-UCSD Bird-200 (Wah et al., 2011), Stanford Car-196 (Khosla et al., 2011), Stanford Dog-120 (Krause et al., 2013), Flower-102 (Nilsback & Zisserman, 2008), and Oxford-IIIT Pet-37 (Parkhi et al., 2012). By default, we restrict the number of unlabelled images per category in $\mathcal{D}^{\text{train}}$ to 3, *i.e.*, $|\mathcal{D}_c^{\text{train}}| = 3$ (randomly sampled from the training split). Additionally, we evaluate under an imbalanced class distribution scenario where $1 \leq |\mathcal{D}_c^{\text{train}}| \leq 10$. Dataset specifics are detailed in the App. A.

**Evaluation Metrics.** Given the unsupervised nature of the task, a one-to-one match between the elements in the ground truth set $\mathcal{C}$ and the candidate set $\widehat{\mathcal{C}}^*$ can not be guaranteed. For this reason, we employ two synergistic metrics: *Clustering Accuracy* (cACC) and *Semantic Similarity* (sACC) to asses performance in FGVR task, following (Conti et al., 2023). cACC evaluates the model's performance in clustering images from the same category together, but does not consider the semantics of the cluster labels. This gap is filled by sACC, which leverages Sentence-BERT (Reimers & Gurevych, 2019) to compare the semantic similarity of the cluster's assigned name with the ground-truth category. sACC also helps gauge the gravity of errors in fine-grained discovery scenarios. For example, clustering a "Seagull" as a "Gull" would yield a higher sACC compared to grouping it to the "Swallow" cluster, given the former mistake is less severe. In summary, the sACC and cACC jointly ensure that the samples in a cluster are not only similar but also possess the correct semantics.

**Implementation Details.** We used BLIP-2 (Li et al., 2023) Flan-T5$_{\text{xxl}}$ as our VQA, ChatGPT (OpenAI, 2022) gpt-3.5-turbo model as our LLM via its public API, and CLIP (Radford et al., 2021) ViT-B/16 as the VLM. The multi-modal fusion hyperparameter and exemplar augmentation times are set to $\alpha = 0.7$ and $K = 10$. We elaborate prompt design detail in App. B.2.

**Compared Methods.** Given that FGVR *without* expert annotations is a relatively new task, except CLEVER (Choudhury et al., 2023), other baselines are not available in the literature. To this end, we introduced several strong baselines: (**i**) CLIP zero-shot transfer using the ground-truth class names as the upper bound (UB) performance, as only experts can possess the knowledge of true class names in $\mathcal{C}$. (**ii**) WordNet baseline that uses the CLIP and a large vocabulary of 119k nouns sourced from the WordNet (Miller, 1995) as class names. (**iii**) BLIP-2 Flan-T5$_{\text{xxl}}$, a VQA baseline, that indentifies the object category by answering a question of the template `"What is the name of the main object in this image?"`. (**iv**) SCD (Han et al., 2023) that first uses non-parametric clustering to group unlabeled images, and then utilizes CLIP to narrow down the unconstrained vocabulary of 119k-noun WordNet and 11k-bird names sourced from Wikipedia. (**v**) CaSED (Conti et al., 2023) uses CLIP to first retrieve captions from a large 54.8-million-entry knowledge base, a subset of PMD (Singh et al., 2022). After parsing the nouns from the captions, CaSED uses CLIP to classify each image. (**vi**) KMeans (Ahmed et al., 2020) on top of CLIP visual features. (**vii**) Sinkhorn-Knopp based parametric clustering in (Caron et al., 2020) using CLIP and DINO (Caron et al., 2021) features. All the baselines we re-run use ViT-B/16 as the CLIP vision encoder. Refer to App. C for details.

### 3.1 BENCHMARKING ON FINE-GRAINED DATASETS

**Quantitative comparison I: The battle of machine-driven approaches.** We benchmarked our FineR system for the task of FGVR without expert knowledge on the five fine-grained datasets. We first evaluated on a minimal training set comprising three random images per category ($|\mathcal{D}_c^{\text{train}}| = 3$). As shown in Tab. 1, our FineR system outperforms the second-best method (BLIP-2) by a substantial margin, giving improvements of $+9.8\%$ in cACC and $+5.7\%$ in sACC averaged on the five datasets. We also evaluated a more realistic scenario, *i.e.,* imbalanced (long-tailed) image distribution for class name discovery ($1 \leq |\mathcal{D}_c^{\text{train}}| \leq 10$) and report the results in Tab. 2. Despite the heightened discovery difficulty posed by imbalanced distribution, our FineR system still consistently outperforms the second-best method, achieving an average improvement of $+7.3\%$ in cACC and $+2.2\%$ in sACC.

| | Bird-200 | | Car-196 | | Dog-120 | | Flower-102 | | Pet-37 | | Average | |
|---|---|---|---|---|---|---|---|---|---|---|---|---|
| | cACC | sACC | cACC | sACC | cACC | sACC | cACC | sACC | cACC | sACC | cACC | sACC |
| Zero-shot (UB) | 57.4 | 80.5 | 63.1 | 66.3 | 56.9 | 75.5 | 69.7 | 77.8 | 81.7 | 87.8 | 65.8 | 77.6 |
| CLIP-Sinkhorn | 23.5 | - | 18.1 | - | 12.6 | - | 30.9 | - | 23.1 | - | 21.6 | - |
| DINO-Sinkhorn | 13.5 | - | 7.4 | - | 11.2 | - | 17.9 | - | 5.2 | - | 19.1 | - |
| KMeans | 36.6 | - | 30.6 | - | 16.4 | - | 66.9 | - | 32.8 | - | 36.7 | - |
| WordNet | 39.3 | 57.7 | 18.3 | 33.3 | 53.9 | **70.6** | 42.1 | 49.8 | 55.4 | 61.9 | 41.8 | 54.7 |
| BLIP-2 | 30.9 | 56.8 | 43.1 | 57.9 | 39.0 | 58.6 | 61.9 | **59.1** | 61.3 | 60.5 | 47.2 | 58.6 |
| CLEVER † | 7.9 | - | - | - | - | - | 6.2 | - | - | - | - | - |
| SCD † | 46.5 | - | - | - | **57.9** | - | - | - | - | - | - | - |
| CaSED | 25.6 | 50.1 | 26.9 | 41.4 | 38.0 | 55.9 | **67.2** | 52.3 | 60.9 | 63.6 | 43.7 | 52.6 |
| FineR (Ours) | **51.1** | **69.5** | **49.2** | **63.5** | 48.1 | 64.9 | 63.8 | 51.3 | **72.9** | **72.4** | **57.0** | **64.3** |

Table 1: cACC(%) and sACC (%) comparison on the five fine-grained datasets. $|\mathcal{D}_c^{\text{train}}| = 3$. Results reported are averaged over 10 runs. †: SCD and CLEVER results are quoted from original paper (SCD uses the *entire* dataset for class name discovery and assumes the number of classes known as *a-priori*). Best and second-best performances are coloured Green and Red, respectively. Gray presents the upper bound (UB).

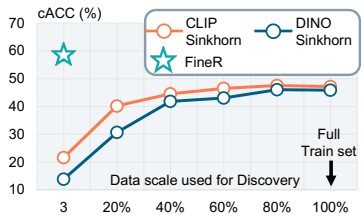

Figure 4: Comparison with the learning-based methods. cACC is averaged on five datasets.

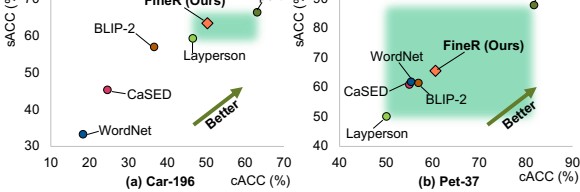

Figure 5: Human study results. Averages computed across 30 participants are reported.

Among the compared methods, BLIP-2 stands out, largely owing to its powerful vision-aligned Flan-T5$_{\text{xxl}}$ language core and its large training knowledge base (Li et al., 2023). The WordNet baseline, SCD, and CaSED show strong performance on specialized datasets such as Dog-120 and Flower-102, largely due to their exhaustive knowledge bases. Specifically, WordNet and SCD cover all ground-truth Dog-120 categories, while CaSED's PMD (Singh et al., 2022) knowledge base includes 101 of 102 ground-truth Flower-102 categories. In contrast, our *reasoning-based* FineR system achieves significant improvements across various datasets *without* explicitly needing to query any external knowledge base. Moreover, with just a few unlabeled images, our method surpasses learning-based approaches that utilize the full-scale training split, as illustrated in Fig. 4.

| Average | cACC | sACC |
|---|---|---|
| Zero-shot (UB) | 65.8 | 77.6 |
| WordNet | 41.8 | 54.7 |
| BLIP-2 | 44.6 | 59.0 |
| CaSED | 40.8 | 51.1 |
| FineR (Ours) | **51.9** | **61.2** |

Table 2: Comparison with *imbalanced* $\mathcal{D}^{\text{train}}$ across five fine-grained datasets. Averages reported.

**Quantitative comparison II: From layperson to expert - where do we stand?** Echoing with our initial motivation of democratizing FGVR, we conducted a human study to establish layperson-level baselines on the Car-196 and Pet-37 datasets. In short, we presented one image per category to 30 non-expert participants and asked them to identify the specific car model or pet breed. If unsure, the participants were asked to describe the objects. The collected answers were then used to build a zero-shot classifier with CLIP, and forms the **Layperson** baseline. For the **Expert** baseline we have used the UB baseline, which uses the ground-truth class names, as described before. As shown in Fig. 5, on the Car-196 dataset, the Layperson baseline outperforms all machine-based methods, except our FineR system. On the Pet-37 dataset, our method distinguishes itself as the top performer among machine-based approaches. This human study shows that FineR successfully narrows the gap between laypersons and experts in FGVR. Further details in App. J.

**Qualitative comparison.** We visualize and analyze the predictions of different methods in Fig. 6. On the Bird-200 dataset (1st row), our FineR system shines in recognizing specific bird species, notably the "Dark-eyed Junco". Our FineR system successfully captures the nuance of the "dark-eyed" visual feature, setting it apart from visually similar birds like the more generic "Junco". In contrast, the compared methods tend to predict coarse-grained and common categories, like "Junco", as they

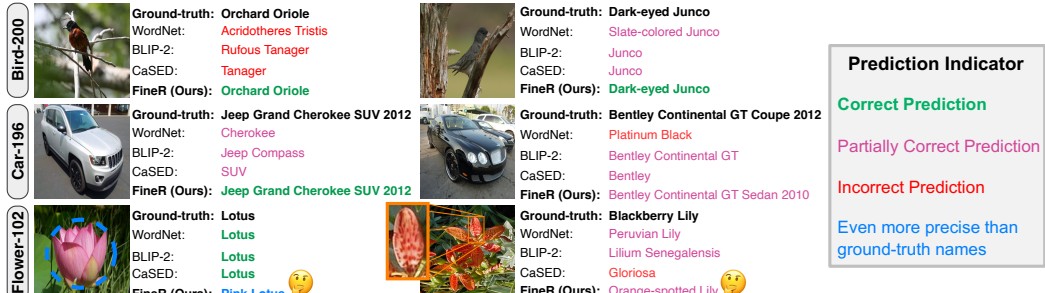

Figure 6: Qualitative comparison on Bird-200, Car-196, and Flower-102 datasets. Digital zoom recommended.

do not emphasize or account for finer details necessary in FGVR. Similar trends are evident in the example of "Jeep Grand Cherokee SUV 2012" (2nd row left). While all methods struggle with the "Bentley Continental GT Couple 2012" (2nd row right), our system offers the closest and most fine-grained prediction. The most striking observation comes from the Flower-102 dataset. Our system outshines the ground-truth in the prediction results of the "Lotus" category (4th row left), classifying it more precisely as a "Pink Lotus" aided by the attribute information "`primary flower color: pink`" during reasoning. And in cases where all models misidentify the "Blackberry Lily" (4th row right), our system offers the most plausible prediction, the "Orange-spotted Lily", informed by the flower's distinctive orange spots in the petals. This further confirms that our system effectively captures fine-grained visual details from images and leverages them for reasoning. This qualitative analysis demonstrates that FineR not only generates precise, fine-grained predictions but also displays high semantic awareness. This holds true even when predictions are only partially correct, thereby mitigating the severity during misclassification. Refer to App. H for more qualitative results.

## 3.2 BENCHMARKING ON THE NOVEL POKEMON DATASET

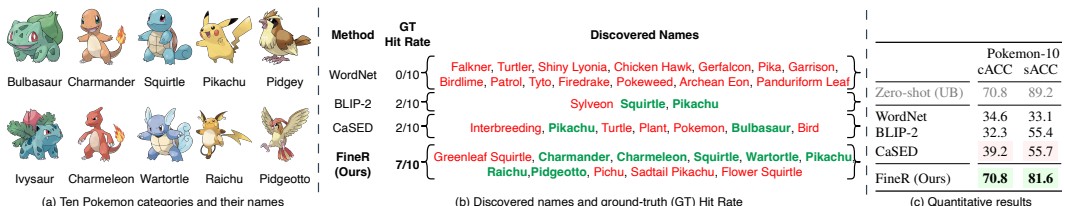

Figure 7: Comparison on the novel Pokemon dataset (3 images per category for discovery, 10 for evaluation).

To further investigate the FGVR capability of FineR on more novel concepts, we introduce a new Pokemon dataset comprised of 10 Pokemon characters, sourced from Pokedex (Nintendo, 2023) and Google Image Search, as shown in Fig. 7(a). One can notice that each pair of Pokemons (each column) have subtle visual differences. As shown in Fig. 7(b), it is hardly surprising that the WordNet baseline fails to discover any of the Pokemon categories, scoring 0/10, given the absence of most specific Pokemon names in its knowledge base. BLIP-2 and CaSED appear to mainly identify only the most common Pokemon classes. Although CaSED does have all ten ground-truth Pokemon names in its PMD knowledge base, it still fails to discover most of these categories. We conjecture this failure to the high visual similarity between the Pokemons characters and their real-world analogs, compounded by CLIP scoring preferences (Ge et al., 2023). As revealed in Fig. 7(b), the classes identified by CaSED predominantly feature real-world categories resembling the Pokemons (*e.g.*, the animal "turtle" rather than the character "Squirtle"). In stark contrast, our FineR system successfully discovers 7/10 ground-truth Pokemon categories, consequently outperforming the second-best result by +31.6% in cACC and +25.9% in sACC as shown in Fig. 7(c).

## 3.3 ABLATION STUDY

We report an ablation analysis of the proposed components of FineR in Tab. 3. As shown in row 2 of Tab. 3, the Noisy Name Denoiser (NND) for the name refinement process (Sec. 2.2.2) stands out as the most impactful, improving cACC by +6.0% and sACC by +4.7% over the baseline that simply uses the preliminary candidate names $\hat{\mathcal{C}}$ for classification. This validates its effectiveness in

filtering out noisy candidate names. Next in row 3 we ablate the MMC (Sec. 2.2.3) by setting $\alpha = 0.7$ and without using $K$ augmentations. We notice an improved clustering accuracy due to the introduction of visual features with the MMC, showing the complementarity of textual and visual classifier. Finally, in row 4 when we run our full pipeline with $K = 10$ we notice a further improved performance for both the metrics. This improvement comes due to the alleviation of limited visual perspectives present in a few images in $\mathcal{D}^{\text{train}}$. Due to lack of space, further ablation study, sensitivity analysis and generalizability study with other LLMs can be found in App. E, App. G and App. F, respectively.

| NND | MMC | $\alpha$ | $K$ | Average | |
|---|---|---|---|---|---|
| | | | | cACC | sACC |
| ✗ | ✗ | ✗ | ✗ | 47.5 | 58.3 |
| ✓ | ✗ | ✗ | ✗ | 53.5 | 63.0 |
| ✓ | ✓ | ✓ | ✗ | 56.9 | 63.3 |
| **✓** | **✓** | **✓** | **✓** | **57.0** | **64.3** |

Table 3: Ablation study on the proposed components. Averages across the five fine-grained datasets are reported.

## 4  RELATED WORK

**Fine-grained visual recognition**. The goal of FGVR (Welinder et al., 2010; Maji et al., 2013; Wei et al., 2021) is to classify subordinate-level categories under a super-category that are visually very similar. Owing to the very similar visual appearances in objects the FGVR methods heavily rely on, aside from ground truth annotations, expert-given auxiliary annotations (Krause et al., 2013; Zhang et al., 2014; Vedaldi et al., 2014; He & Peng, 2017). The FGVR methods can be categorized into two types: (i) feature-encoding methods (Lin et al., 2015; Gao et al., 2016; Yu et al., 2018; Zheng et al., 2019) that focus on extracting effective features (*e.g.*, using bilinear pooling) for improved recognition; and (ii) localization methods (Huang et al., 2016; Fu et al., 2017; Simonelli et al., 2018; Huang & Li, 2020) instead focus on selecting the most discriminative regions for extracting features. Moreover, TransHP (Wang et al., 2023) learns soft text prompts for coarser-grained information, aiding fine-grained discrimination, while V2L (Wang et al., 2022) combines vision and language models for superior FGVR performance. Unlike the aforementioned methods, which rely on expert-defined fine-grained categories, our approach operates in a *vocabulary-free* manner. Recently, in an attempt to make FGVR annotation free, CLEVER (Choudhury et al., 2023) was proposed that first extracts non-expert image descriptions from images and then trains a fine-grained textual similarity model to match descriptions with wikipedia documents at a sentence-level. Our FineR is similar in spirit to CLEVER, except we exploit LLMs to reason about category names from visual descriptions and classify the fine-grained classes without the need of training any module. An additional advantage of FineR over the work in (Choudhury et al., 2023) is that FineR can work in low data regime.

**LLM enhanced visual recognition**. In the wake of foundation models, commonly addressed tasks such as image classification and VQA have benefited with the coupling of vision with language, either by means of LLMs (Menon & Vondrick, 2023; Yang et al., 2022; Yan et al., 2023) or querying external knowledge-base (Conti et al., 2023; Han et al., 2023). Specifically, some recent image classification approaches (Pratt et al., 2023; Menon & Vondrick, 2023; Lin et al., 2023) first decompose class names into more descriptive captions using GPT-3 (Brown et al., 2020) and then use CLIP (Radford et al., 2021) to classify the images. In a similar manner, recent VQA approaches (Yang et al., 2022; Shao et al., 2023; Berrios et al., 2023) first extract captions using off-the-shelf captioning model and then feeds the question, caption and in-context examples to induce GPT-3 to arrive at the correct answer. However, unlike FineR, none of the approaches have been explicitly designed to work in the vocabulary-free fine-grained visual recognition setting.

## 5  CONCLUSION

In this work we showed the expert knowledge is no longer a roadblock to build effective FGVR systems. In detail, we presented FineR that exploits the world knowledge inherent to the LLMs to address FGVR, thereby disposing off the need of requiring expert annotations. To make LLMs compatible with a vision-based task we first translate essential visual attributes and their descriptions from images into text using non-expert VQA models. Through well-structured prompts and visual attributes we invoke the reasoning ability of LLMs to discover candidate class names in the dataset. Our training-free approach has been evaluated on multiple FGVR benchmarks, and we showed that it can outperform several state-of-the-art VQA, FGVR and learning-based approaches. Additionally, through the evaluation on a newly collected dataset we demonstrated the versatility of FineR in working in the wild where expert annotations are hard to obtain.

## 6 ACKNOWLEDGMENTS

This work was supported by the MUR PNRR project FAIR - Future AI Research (PE00000013) , funded by the NextGenerationEU. E.R. is partially supported by the PRECRISIS, funded by the EU Internal Security Fund (ISFP-2022-TFI-AG-PROTECT-02-101100539), the EU project SPRING (No. 871245), and by the PRIN project LEGO-AI (Prot. 2020TA3K9N). This work was carried out in the Vision and Learning joint laboratory of FBK and UNITN. We extend our appreciation to Riccardo Volpi and Tyler L. Hayes for their helpful discussions regarding dataset licensing. Furthermore, we are deeply thankful to Gabriela Csurka and Yannis Kalantidis for their valuable suggestions during the discussion stage.

## 7 ETHICS STATEMENT

Large pre-trained models such as CLIP (Radford et al., 2021), BLIP-2 (Li et al., 2023), and Chat-GPT (OpenAI, 2022) inherit biases from their extensive training data (Goh et al., 2021; Schramowski et al., 2022; Kirk et al., 2021). Given that FineR incorporates these models as core components, it too is susceptible to the same biases and limitations. Consequently, model interpretability and transparency are central to our design considerations. Owing to the modular design of the proposed FineR system, every intermediate output within the system's scope is not only interpretable but also transparent, facilitating traceable final predictions. For instance, as demonstrated in the last row of Fig. 6 in the main paper, when our system produces a more accurate prediction like `"Pink Lotus"` compared to the ground-truth label, this prediction can be directly explained by tracing back to its attribute description pair `"primary flower color: pink"`. Similarly, for a partially accurate prediction like `"Orange-spotted Lily"`, the system's rationale can be traced back to the attribute description pair `"petal color pattern: orange-spotted"`. The semantically meaningful output of our model also mitigates the severity of incorrect predictions. Through this approach, we hope to offer a system that is both interpretable and transparent, countering the inherent biases present in large pre-trained models. In addition, in relation to our human study, all participants were volunteers and received no financial compensation for their participation. All participants were informed about the study's objectives and provided written informed consent before participating. Data were anonymized to protect participants' privacy, and no personally identifiable information was collected. Participants were free to withdraw themselves or their responses from the study at any time.

## 8 REPRODUCIBILITY STATEMENT

Our FineR system is training-free and built on openly accessible models: BLIP-2 (Li et al., 2023) for visual question answering model, CLIP (Radford et al., 2021) for visuan language model, and ChatGPT gpt-3.5-turbo (OpenAI, 2022) for large language model. All models are publicly accessible and do not require further fine-tuning for utilization. BLIP-2 and CLIP offer publicly accessible weights, while the ChatGPT gpt-3.5-turbo model is available through a public API. We confirm the generalizability to open-source large language models by replacing the large language model core with Vicuna-13B (Chiang et al., 2023), as studied in App. F. Comprehensive details for replicating our system are outlined in Sec. 2, with additional details about prompts provided in App. B.2. Upon publication, we will release all essential resources for reproducing the main experimental results in the main paper, including code, prompts, datasets and the data splits. To enable seamless replication of our primary experimental results from any intermediate stage of our FineR system, we will also release the intermediate outputs from each component. These intermediate outputs include the super-category names identified by BLIP-2, attribute names and descriptions derived from both the large language model and BLIP-2, as well as reasoning results given by the large language model for each dataset. This comprehensive approach ensures the reproducibility of our proposed method and the results, and aims to provide practitioners with actionable insights for performance improvement.

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

# APPENDIX

In this Appendix, we offer a thorough set of supplementary materials to enrich the understanding of this work. We commence by delving into the specifics of the five fine-grained datasets, as well as our newly introduced Pokemon dataset, in App. A. Following this, App. B elaborates on the implementation details and prompt designs integral to our FineR system. App. C outlines the implementation details for the compared methods in our experiments. Supplementary experimental results for the imbalanced image distribution and the component-wise ablation study are covered in Sec. D and App. E, respectively. We expand on the generalizability of our approach with open-source LLMs in App. F. App. G examines the sensitivity of our system to different factors. Additional qualitative results are disclosed in App. H. Additionally, a qualitative analysis of failure cases is provided in App. I. Our human study design is discussed in App. J. Lastly, this Appendix concludes with further insights, intriguing discussions, and setting comparison App. K.

## A DATASET DETAILS

### A.1 FIVE STANDARD FINE-GRAINED DATASETS

| | | Bird-200 | Car-196 | Dog-120 | Flower-102 | Pet-37 |
|---|---|---|---|---|---|---|
| $\|\mathcal{D}_c^{\text{train}}\| = 3$ | $\|\mathcal{C}\|$ | 200 | 196 | 120 | 102 | 37 |
| | $\|\mathcal{D}^{\text{train}}\|$ | 600 | 588 | 360 | 306 | 111 |
| | $\|\mathcal{D}^{\text{test}}\|$ | 5.7k | 8.1k | 8.6k | 2.0k | 3.7k |
| $1 \leq \|\mathcal{D}_c^{\text{train}}\| \leq 10$ | $\|\mathcal{C}\|$ | 200 | 196 | 120 | 102 | 37 |
| | $\|\mathcal{D}^{\text{train}}\|$ | 438 | 458 | 274 | 229 | 115 |
| | $\|\mathcal{D}^{\text{test}}\|$ | 5.7k | 8.1k | 8.6k | 2.0k | 3.7k |

Table 4: Statistics for the five fine-grained datasets. The number of categories $|\mathcal{C}|$, the number of images used for discoveries $|\mathcal{D}^{\text{train}}|$, and the number of images used for test $|\mathcal{D}^{\text{test}}|$ are reported for the two cases where 3 samples per category are used for discovery ($|\mathcal{D}_c^{\text{train}}| = 3$, see upper half), and few arbitrary images are used for discovery ($1 \leq |\mathcal{D}_c^{\text{train}}| \leq 10$, see lower half), respectively.

We provide comprehensive statistics for the five well-studied fine-grained datasets: Caltech-UCSD Bird-200 (Wah et al., 2011), Stanford Car-196 (Khosla et al., 2011), Stanford Dog-120 (Krause et al., 2013), Flower-102 (Nilsback & Zisserman, 2008), and Oxford-IIIT Pet-37 (Parkhi et al., 2012), used for our experiments in Tab. 4. To the Car-196 dataset, we remove all the license plate numbers from images featuring specific plates.

In the default experimental setting, where each category has 3 images available for class name discovery ($|\mathcal{D}_c^{\text{train}}| = 3$), we randomly select 3 images per category from the training split of each dataset to constitute the few unlabeled sample set $\mathcal{D}^{\text{train}}$. The test split serve as the evaluation set $\mathcal{D}^{\text{test}}$. It is ensured that the discovery and test sets are mutually exclusive, $\mathcal{D}^{\text{train}} \cap \mathcal{D}^{\text{test}} = \emptyset$. To evaluate our method under more realistic and challenging conditions, we construct an imbalanced discovery set and report the evaluation results in Tab. 3. This set features not only a restricted number of images per category but also conforms to a synthetic long-tail distribution. To generate this imbalanced set, we first generate a long-tail distribution, where each category contains between 1 and 10 images, using Zipf's law (Newman, 2005) with a shape parameter of 2.0. We then use this distribution to randomly sample the corresponding amount of images per category from each dataset, forming the imbalanced discovery set $\mathcal{D}^{\text{train}}$. Fig. 8 displays the long-tail distribution we employ across categories in the five fine-grained datasets.

### A.2 POKEMON DATASET

Beyond the five standard fine-grained datasets, we build and introduce a Pokemon dataset of virtual objects to further evaluate both the baselines and our FineR system. Initially, we manually select 10 visually similar Pokemon categories by checking the Pokedex[1], an official Pokemon index. We then use Google Image Search to collect 13 representative images for each category, each of which is then manually verified and annotated. In alignment with our previous experimental settings, we designate 3 images per category ($|\mathcal{D}_c^{\text{train}}| = 3$) to form the discovery set $\mathcal{D}^{\text{train}}$, while the remaining images are allocated to the test set $\mathcal{D}^{\text{test}}$. As the Pokemon dataset contains only 130 images in total, we directly illustrate the dataset and its subdivisions in Fig. 9. Despite its limited size, this dataset presents multiple challenges: (i) The dataset is fine-grained; for instance, the primary visual

---

[1]https://www.pokemon.com/us/pokedex

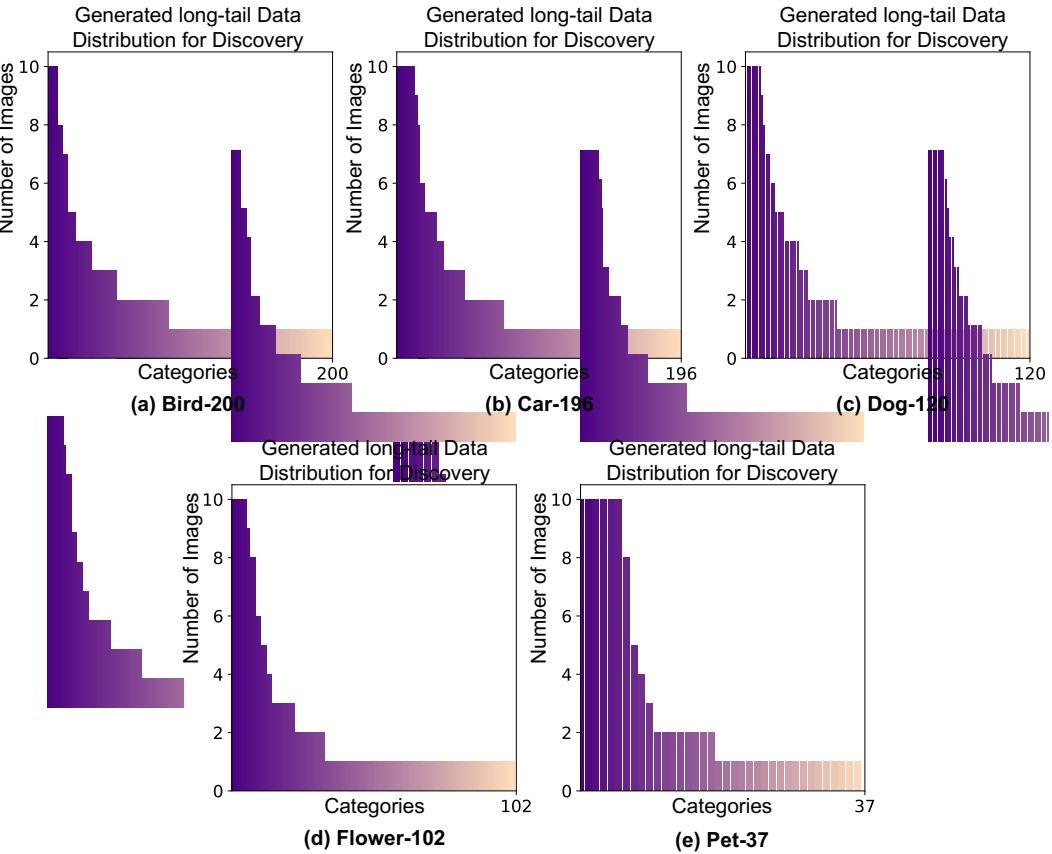

Figure 8: Imbalanced discovery set ($1 \le |\mathcal{D}_c^{\text{train}}| \le 10$) histograms of the number of unlabeled samples per category following the generated long-tail distribution for the five fine-grained datasets, respectively.

difference between `"Bulbasaur"` and `"Ivysaur"` is a blooming flower on `"Ivysaur"`'s back; (ii) Test images might contain multiple objects. (iii) A domain gap exists compared to real-world photos and physical entities. We plan to publicly release this Pokemon dataset along with our code.

## B   IMPLEMENTATION DETAILS OF FINER

### B.1   DATA AUGMENTATION PIPELINE

To mitigate the strong bias arising from the limited samples in $\mathcal{D}^{\text{train}}$ during the construction of the multi-modal classifier, we employ a data augmentation strategy. Each sample $x \in \mathcal{D}^{\text{train}}$ is transformed $K$ times by a random data augmentation pipeline. Therefore, $K$ augmented samples are generated. This approach is crucial for preparing the system for real-world scenarios where test images may feature incomplete objects, varied color patterns, and differing orientations or perspectives. Accordingly, our data augmentation pipeline incorporates random cropping, color jittering, flipping, rotation, and perspective shifts. Additionally, we integrate random choice and random apply functions within the pipeline to more closely emulate real-world data distributions.

### B.2   PROMPT DESIGN

In this section, we delves into the prompt designs of FineR. BLIP-2 is used as the VQA model to construct the Visual Information Extractor (VIE) in FineR system. ChatGPT (OpenAI, 2022) gpt-3.5-turbo is used as the frozen LLM in our system, with temperature of 0.9 to purse diversity for fine-grained categories. In the subsequent sub-sections, we employ the reasoning process for an image of an `"Italian Greyhound"` as an intuitive example to facilitate our discussion on prompt designs.

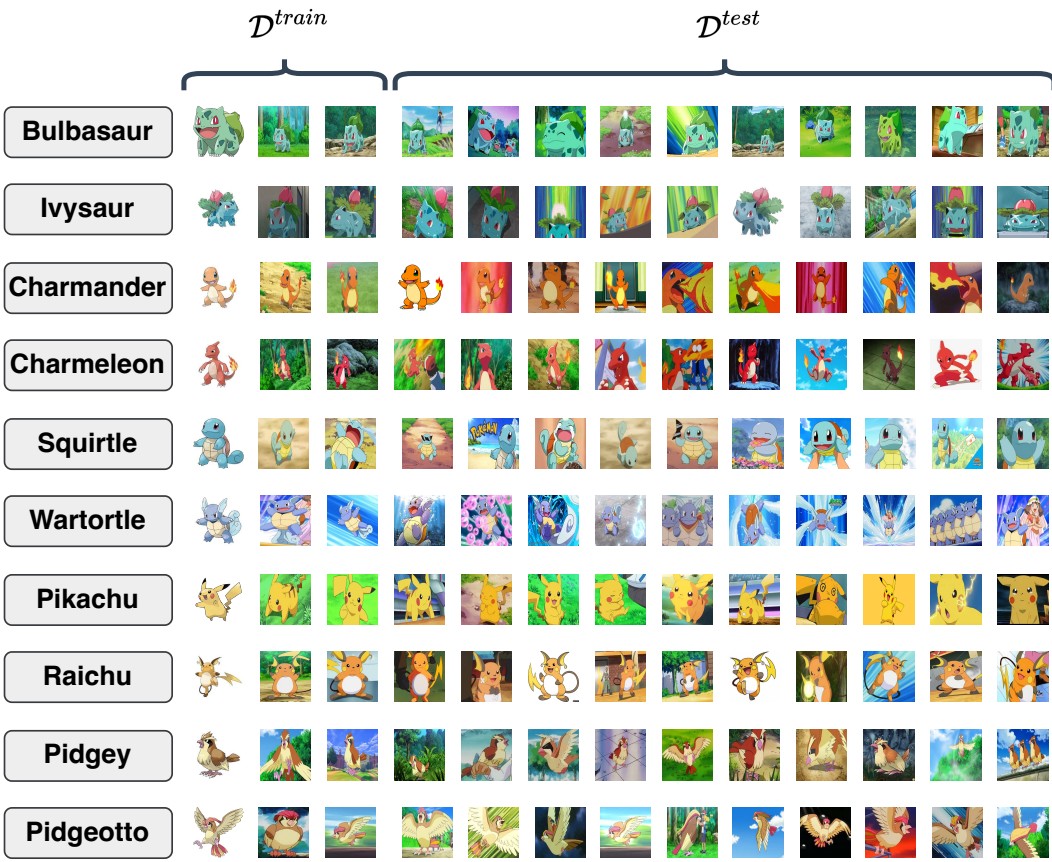

Figure 9: Visualization of the newly introduced Pokemon dataset. The images in the first three columns at each row are used for discovery.

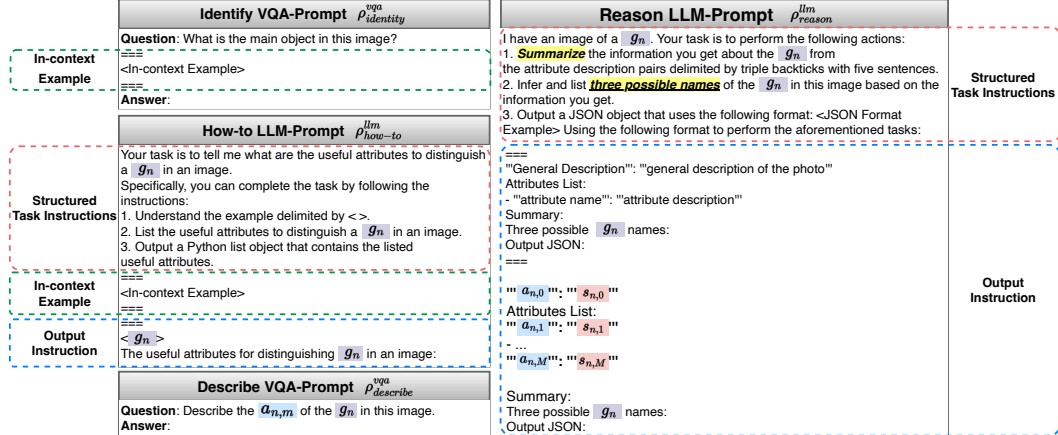

Figure 10: The templates of the prompts used in FineR.

Fig. 10 presents the prompt templates used in each module of our FineR system, and Fig. 11 provides a detailed illustration of the entire reasoning process, including the *exact* outputs at each step.

**Identify VQA-Prompt.** Since language models are unable to *see*, the initial crucial step is to translate pertinent visual information into textual modality, serving as visual cues for the language model. As illustrated in Fig. 2, our preliminary research indicates that, while VQA methods may struggle to pinpoint fine-grained semantic categories due to their limited training knowledge, they show robust capability at identifying coarse-grained super-categories like "Bird", "Dog", and even "Pokemon". As an initial step, we employ our VQA-based Visual Information Extractor (VIE) to

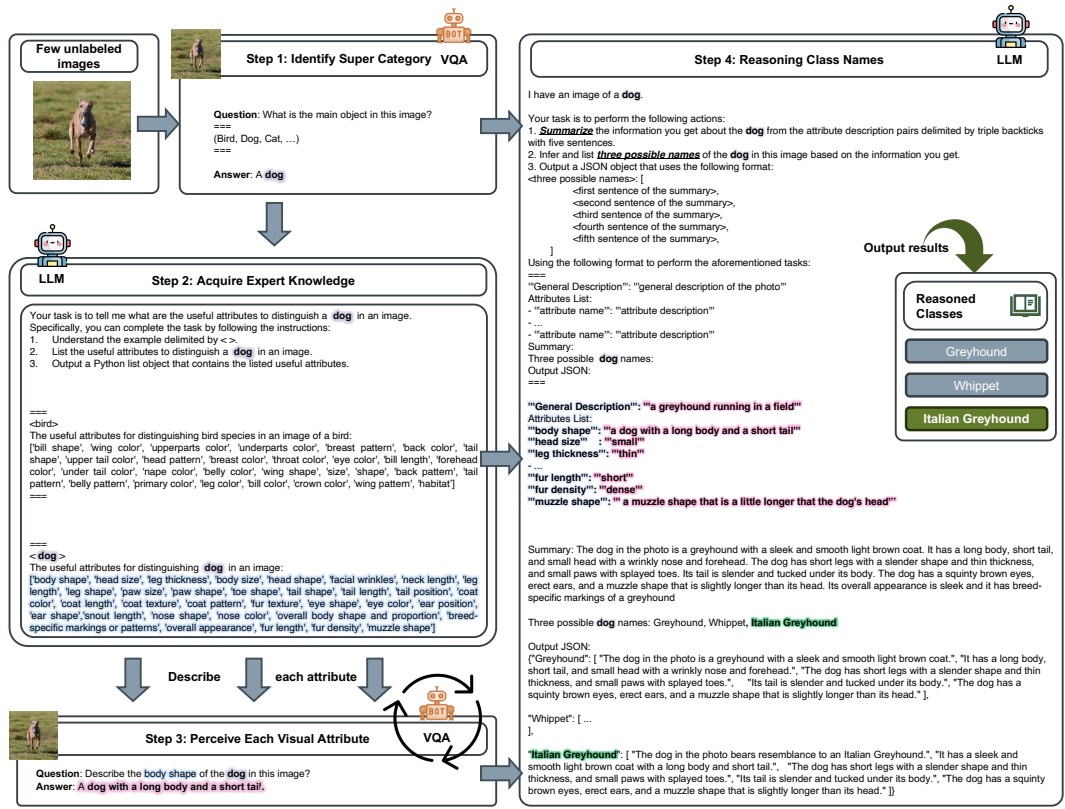

Figure 11: **Whole category reasoning pipeline of FineR**. An image of a `"Italian Greyhound"` is used as an example.

initially identify these super-categories from the input images. These identified super-categories then serve as coarse-grained base information to direct subsequent operations. Importantly, this design ensures that our FineR system remains dataset-agnostic; we do not presume any super-category affiliation for a target dataset *a-priori*. To steer the VIE towards recognizing super-categories, we include a simple in-context example (Brown et al., 2020) within the **Identify VQA-prompt**, which conditions the granularity of the output. The specific in-context example utilized in $\rho_{\text{identify}}^{\text{vqa}}$ is shown in Fig. 12. The VIE module successfully identifies the super-category `"dog"` from the image of a `"Italian Greyhound"`. We find that providing straightforward in-context examples, such as `"Bird, Dog, Cat, ..."`, effectively guides the VQA model to output super-categories like `"car"` or `"flower"` for the primary object in an image. We use the same in-context example for *all* the datasets and experiments. Users can also effortlessly adjust this in-context example to better align the prompt with their specific data distribution.

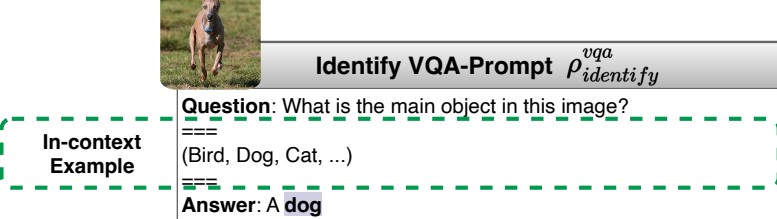

Figure 12: **Identify VQA-prompt** details. An image of a `"Italian Greyhound"` is used as an example.

**How-to LLM-Prompt.** After the super-category names $\mathcal{G}$ (e.g., `"dog"`)is obtained, we can then invoke the Acquire Expert Knowledge (AEK) module to acquire useful visual attributes for distinguishing the fine-grained sub-categories of $\mathcal{G}$, using the LLM. To give a sense of what are the *useful* attributes, we add an in-context example of bird using the visual attributes collected by (Wah et al., 2011) due to its availability, as shown in Fig. 13. This bird attributes in-context example is used for *all* datasets. Although we used the attribute names of birds as an in-context example, we also tried a version where we do not use bird attributes in the in-context example and we get similar results.

The in-context example of birds can guide LLM to give cleaner answers and thereby simplify the post-parsing process. Next, we embed the super-category identified from the previous step into the **How-to LLM-prompt** and query the LLM to acquire the useful visual attributes as expert knowledge to help the following steps. In addition, we also specify a output format in the task instructions for the ease of automatic processing. To obtain as diverse a variety of useful visual attributes as possible, here the AEK module query a LLM 10 times and use the union of the output attributes for the following process with a temperature of 0.9 for higher diversity. As detailed in Fig. 13, after ten iterations of querying, we amass a set of valuable attributes useful for distinguishing a "dog".

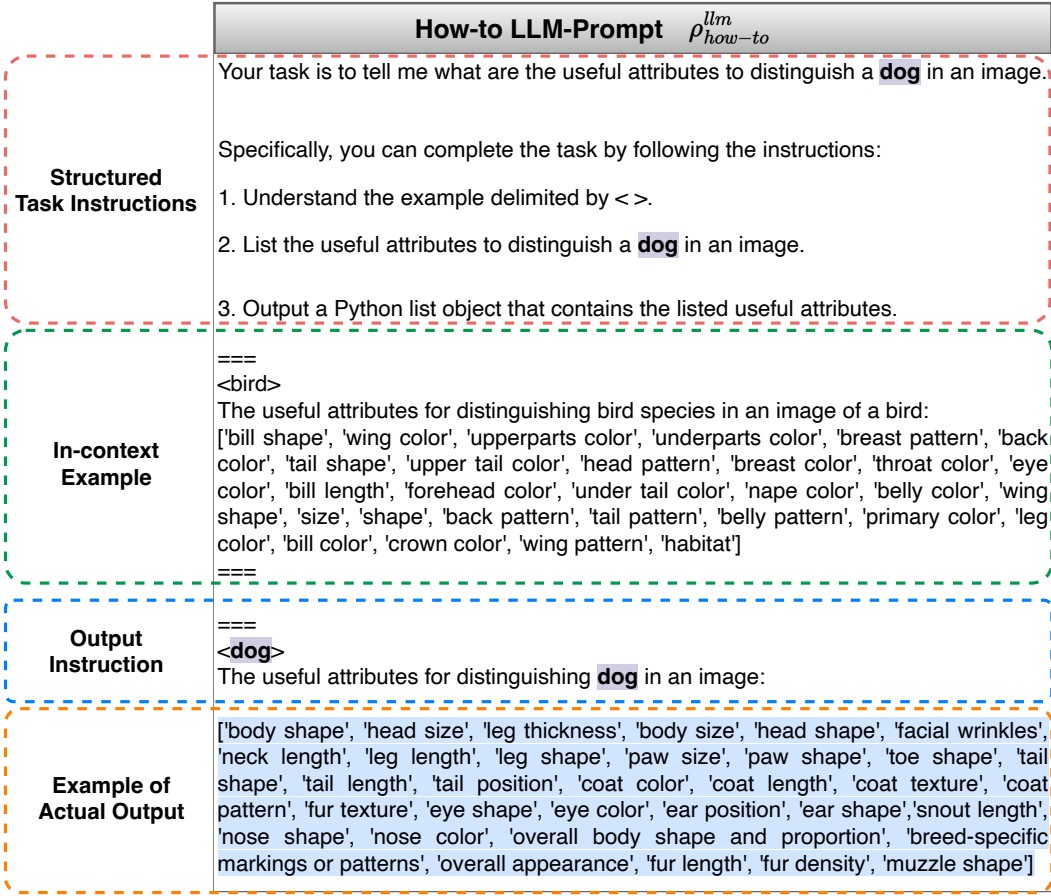

Figure 13: **How-to LLM-prompt** details. "dog" is used as an example.

**Describe VQA-Prompt.** Having the acquired useful attributes set $\mathcal{A}$ as expert knowledge, we can then extract visual information targeted by the attributes from the given image using our VIE module. A simple "Question: Describe the $a_{n,m}$ of the $g_n$ in this image. Answer:" VQA-prompt is used iteratively to query the VQA to describe each attribute $a_{n,m}$ for the corresponding super-category $g_n$. As shown in Fig. 14, a description $s_{n,m}$ (*e.g.*, "a dog with a long body and a short tail") is obtained for the visual attribute $a_{n,m}$ (*e.g.*, "body shape").

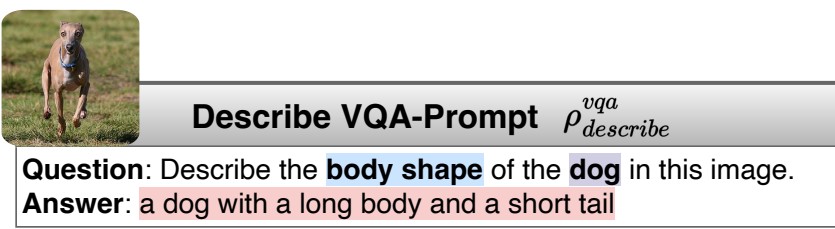

Figure 14: **Describe VQA-prompt** details. An image of a "Italian Greyhound" is used as an example.

**Reason LLM-Prompt.** At this stage, we have acquired super-category-attribute-description triplets, denoted as $\mathcal{G}$, $\mathcal{A}$, $\mathcal{S}$, enriched with visual cues essential for reasoning about fine-grained categories. These cues serve as a prompt to engage the large language model core of our Semantic

Category Reasoner (SCR) in fine-grained semantic categories reasoning. The **Reason LLM-prompt**, illustrated in Fig. 15, comprises two main sections: (i) structured task instructions; (ii) output instruction. Key design elements warrant special mention. Firstly, introducing the sub-task "1. Summarize the information you get about the $g_n$ from the attribute-description pairs delimited by triple backticks with five sentences" enables the language model to "digest" and "refine" incoming visual information. Inspired by zero-shot Chain-of-Thought (CoT) prompting (Kojima et al., 2022) and human behavior, this step acts as a *self-reflective* reasoning mechanism. For example, when kids describes a "bird" they observe to an expert, the expert typically summarizes the fragmented input before providing any conclusions, thereby facilitating more structured reasoning. A subtle but impactful design choice, seen in Fig. 15, is to include the summary sentences in the output JSON object, even if these are not utilized in subsequent steps. This aids in reasoned category names that better align with the summarized visual cues. Secondly, as highlighted in Sec. 2.2.2, we prompt the LLM to reason "three possible names" instead of just one. This approach encourages the LLM to consider a broader range of fine-grained categories. By incorporating these sub-task directives, we construct a zero-shot CoT-like LLM-prompt specifically designed for fine-grained reasoning from visual cues, aligning with the "Let's think step by step" paradigm (Kojima et al., 2022). We also incorporate a task execution template in the output instruction section to explicitly delineate both the input data and output guidelines, enabling the LLM to systematically execute the sub-tasks. Lastly, we specify a JSON output format to facilitate automated parsing. This allows for the effortless extraction of reasoned candidate class names from the resulting JSON output. As illustrated in Fig. 15, the actual reasoning process for an image of a "Italian Greyhound" is elaborated. Of the three possible reasoned names, the correct category is accurately discovered.

## C    IMPLEMENTATION DETAILS OF THE COMPARED METHODS

**KMeans clustering.**    To benchmark clustering accuracy (Roy et al., 2022), we construct a KMeans (Ahmed et al., 2020) baseline utilizing CLIP (Radford et al., 2021) features. For an equitable comparison, we use the unlabeled samples $\mathcal{D}^{\text{train}}$ for cluster formation. Despite its simplicity, KMeans demonstrates competitiveness when empowered by robust features, as evidenced in SCD (Han et al., 2023) and Tab. 1. Importantly, the KMeans baseline presumes prior knowledge about the number of categories in $\mathcal{D}^{\text{train}}$, while our method operates without any such assumptions in all experiments.

**Learning-based Sinkhorn clustering.**    In addition to the KMeans baseline, we introduce two learning-based baselines that leverage features from CLIP and DINO (Caron et al., 2021), using Sinkhorn clustering (Caron et al., 2020). These baselines utilize the Sinkhorn-Knopp algorithm for cross-view pseudo-labeling and a frozen CLIP or DINO ViT-B/16 model for feature extraction. They learn a clustering classifier by optimizing a *swapped* prediction problem. Consistent with other methods in our comparison, we train these baselines on $\mathcal{D}^{\text{train}}$ for 200 epochs, strictly adhering to the hyperparameters specified in the original papers.

**CLEVER.** Curious Layperson-to-Expert Visual Entity Recognition (CLEVER) (Choudhury et al., 2023) is a pioneering work focused on fine-grained visual recognition without expert annotations. It employs web encyclopedias to train a model for generating image captions that highlight visual appearance. A subsequent fine-grained textual similarity model is trained to match these captions with corresponding documents (category), achieving vocabulary-free cross-modal classification.

**SCD.** Semantic Category Discovery (SCD) (Han et al., 2023) is a two-step approach for vocabulary-free zero-shot classification, integrating VLM models with a knowledge base that includes 119k WordNet nouns and 11k bird names from Wikipedia. Initially, SCD leverages KMeans clustering atop DINO features (Caron et al., 2021; Pu et al., 2024; 2023) to categorize images. Subsequently, it utilizes CLIP VLM to extract a set of candidate class names for each cluster from its knowledge bases. An voting mechanism within each cluster refines this set, aiming to pinpoint the most likely candidate names. This refinement process is formed as a linear assignment problem between candidate names and clusters and performed iteratively to optimize the selection of class names. Since the code for SCD is not publicly available, we reference its experimental results from the original paper. It is important to note that SCD utilizes the *entire* dataset for class name discovery and assumes the number of categories to be discovered is known a priori. In contrast, our FineR system relies on only

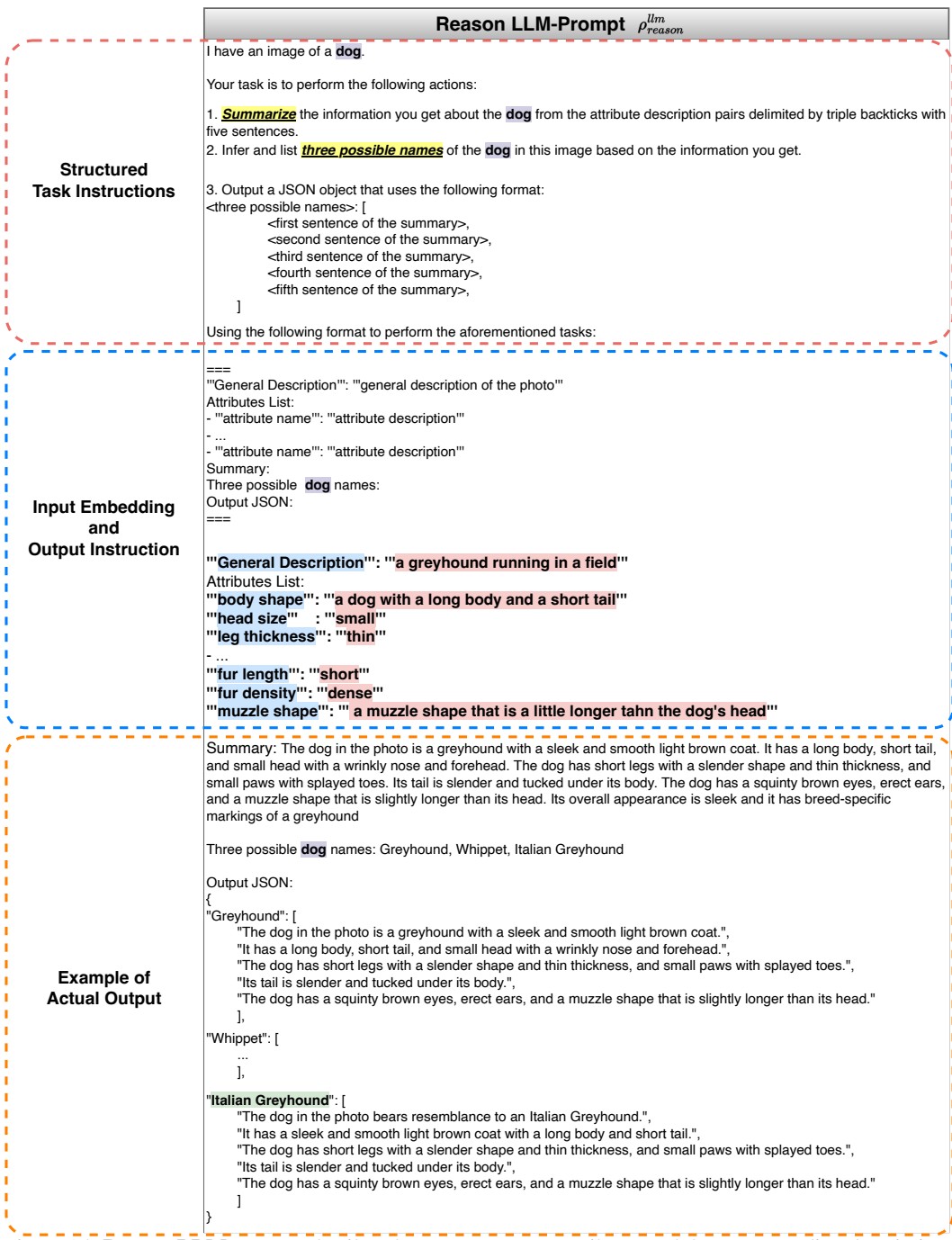

Figure 15: **Reason LLM-prompt** details. The super-category, attributes and the corresponding descriptions acquired from the image of a `"Italian Greyhound"` is used as an example.

a few unlabeled samples for class name identification and does not require any prior knowledge about the number of categories.

**CaSED.** Category Search from External Databases (CaSED) employs CLIP and an expansive 54.8-million-entry knowledge base to achieve vocabulary-free classification. CaSED constructs its extensive knowledge base from the five largest sub-sets of PMD (Singh et al., 2022). It uses CLIP to retrieve a set of candidate class names from its knowledge base based on the vision-language similarity score in CLIP space. We compare with CaSED under our setting. $\mathcal{D}^{\text{train}}$ with CLIP ViT-B/16 are used for class name discovery.

# D    FURTHER COMPARISON WITH IMBALANCED IMAGE DISTRIBUTION

| | Bird-200 | | Car-196 | | Dog-120 | | Flower-102 | | Pet-37 | | Average | |
|---|---|---|---|---|---|---|---|---|---|---|---|---|
| | cACC | sACC | cACC | sACC | cACC | sACC | cACC | sACC | cACC | sACC | cACC | sACC |
| Zero-shot (UB) | 57.4 | 80.5 | 63.1 | 66.3 | 56.9 | 75.5 | 69.7 | 77.8 | 81.7 | 87.8 | 65.8 | 77.6 |
| WordNet | 39.3 | 57.7 | 18.3 | 33.3 | **53.9** | **70.6** | 42.1 | 49.8 | 55.4 | 61.9 | 41.8 | 54.7 |
| BLIP-2 | 27.5 | 56.4 | 38.6 | 57.3 | 36.6 | 57.7 | 58.2 | **58.4** | 62.3 | 65.1 | 44.6 | 59.0 |
| CaSED | 23.5 | 48.8 | 24.6 | 40.9 | 37.8 | 55.4 | **64.8** | 50.2 | 53.0 | 60.2 | 40.8 | 51.1 |
| FineR (Ours) | **46.2** | **66.6** | **48.5** | **62.9** | 42.9 | 61.4 | 58.5 | 48.2 | **63.4** | **67.0** | **51.9** | **61.2** |

Table 5: cACC (%) and sACC (%) comparison on the five fine-grained datasets with *imbalanced* (long-tailed) data ($1 \leq |\mathcal{D}_c^{\text{train}}| \leq 10$) for discovery. Results reported are averaged over 10 runs. Best and second-best performances are coloured Green and Red, respectively. Gray presents the upper bound (UB).

In Tab. 5, we analyze the impact of imbalanced data distribution on class name discovery across five fine-grained datasets. The imbalanced data notably degrades the performance of almost all the methods, making it challenging to identify rare classes. In extreme scenarios, some categories have only a single image available for name discovery. Despite the lower overall performance, the fluctuation in results for each method remains consistent, as detailed in Tab. 5. Nevertheless, our FineR system consistently outperforms other methods, achieving an average improvement of $+7.3\%$ in cACC and $+2.2\%$ in sACC.

# E    FURTHER ABLATION STUDY

| NND | MMC | $\alpha$ | $K$ | Bird-200 | | Car-196 | | Dog-120 | | Flower-102 | | Pet-37 | | Average | |
|---|---|---|---|---|---|---|---|---|---|---|---|---|---|---|---|
| | | | | cACC | sACC | cACC | sACC | cACC | sACC | cACC | sACC | cACC | sACC | cACC | sACC |
| ✗ | ✗ | ✗ | ✗ | 44.8 | 64.5 | 33.8 | 52.9 | 49.7 | 62.8 | 45.4 | 47.6 | 63.8 | 64.0 | 47.5 | 58.3 |
| ✓ | ✗ | ✗ | ✗ | 48.3 | 70.2 | 47.4 | 63.1 | **50.8** | **66.5** | 52.2 | 48.9 | 69.1 | 66.1 | 53.5 | 63.0 |
| ✓ | ✓ | ✗ | ✗ | 49.4 | 68.3 | 45.7 | 62.9 | 39.4 | 61.4 | 68.7 | 50.8 | 65.4 | 65.0 | 53.7 | 61.7 |
| ✓ | ✓ | ✓ | ✗ | 50.4 | 70.0 | 48.9 | 63.3 | 47.3 | 64.8 | 66.1 | 51.1 | 71.8 | 67.3 | 56.9 | 63.3 |
| ✓ | ✓ | ✗ | ✓ | 49.9 | 68.6 | 46.4 | 63.1 | 41.6 | 62.6 | **69.4** | 50.6 | 65.8 | 68.7 | 54.6 | 62.7 |
| ✓ | ✓ | ✓ | ✓ | **51.1** | **69.5** | **49.2** | **63.5** | 48.1 | 64.9 | 63.8 | **51.3** | **72.9** | **72.4** | **57.0** | **64.3** |

Table 6: Ablation study on the proposed components across the five fine-grained datasets. Our full system is colored in Green.

In Sec. 3.3, we validate the effectiveness of individual components in our FineR system. We further present the detailed ablation results in Tab. 6 for the five studied datasets. Consistent with the previous observation, our Noisy Name Denoiser (NND) component improves our system performance across all datasets (second row vs first row), validating its role in noisy name refinement. On the Dog-120 dataset, however, certain enhancements like the multi-modal classifier (MMC) do not contribute to better performance (third row vs second row). This is ascribed to the large visual bias in the samples for discovery, a trend also observed in the Car-196 and Pet-37 datasets (third row vs second row). For instance, the color patterns within the same breed of dogs or cats, or within the same car model, can vary significantly. Despite this, modality fusion hyperparameter $\alpha = 0.7$ and sample augmentation ($K = 10$) partially mitigate this issue on the Car-196 and Pet-37 datasets. Overall, our complete FineR system delivers the best performance. Addressing visual biases stemming from high intra-class variance in fine-grained categories presents an intriguing avenue for future research.

# F    FURTHER GENERALIZABILITY STUDY WITH OPEN-SOURCE LLMS

We replace the LLM of our FineR system with Vicuna-13B (Chiang et al., 2023), an open-source LLM based on LLaMA (Touvron et al., 2023), and evaluate it on the three most challenging fine-grained datasets. The findings, presented in Tab. 7, reveal that the performance of our system remains consistent whether using Vicuna-13B or ChatGPT for the FGVR task. Both language models are effectively leveraged by FineR to reason fine-grained categories based on the provided visual cues. These results suggest that our FineR system is robust and generalizable across different LLMs.

| | Bird-200 | | Car-196 | | Dog-120 | |
| | cACC | sACC | cACC | sACC | cACC | sACC |
|---|---|---|---|---|---|---|
| Vicuna-13B | 50.7 | 68.9 | 47.9 | 61.7 | **50.3** | **67.2** |
| ChatGPT | **51.1** | **69.5** | **49.2** | **63.5** | 48.1 | 64.9 |

Table 7: Generalization study with open-source LLM Vicuna-13B.

## G FURTHER SENSITIVITY ANALYSIS

In this section, we extend our component ablation study to scrutinize the sensitivity of our FineR system to various factors. We explore the impact of the hyperparameter $\alpha$ on multi-modal fusion during classifier construction in App. G.1. Additionally, we examine the effects of sample augmentation times $K$ in mitigating visual bias due to limited sample sizes in App. G.2. We also analyze the system's performance under varying amounts of unlabeled images per category, denoted by $\mathcal{D}_c^{\text{train}}$, for class name discovery. Finally, we assess the influence of CLIP VLM model size on our system performance. Clustering Accuracy (cACC) and Semantic Similarity (sACC) are reported across all five fine-grained datasets.

### G.1 ANALYSIS OF THE HYPERPARAMETER $\alpha$

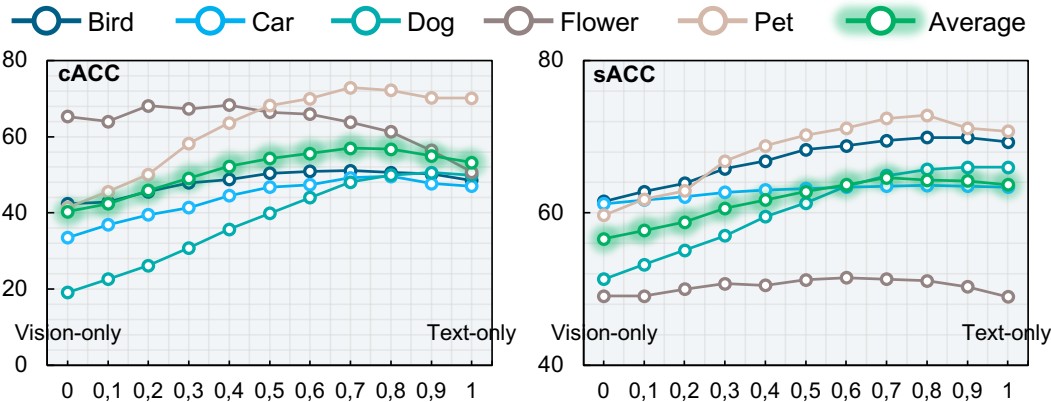

Figure 16: Sensitivity analysis of $\alpha$ in FineR system. The average performance is highlighted in **Green**.

The hyperparameter $\alpha$ modulates multi-modal fusion in our classifier, enhancing performance when coupled with sample augmentation, as discussed in Sec. 3.3. Ranging from 0 to 1, $\alpha$ balances visual and textual information. As shown in Fig. 16, we observe an optimal trade-off at $\alpha = 0.7$ across most datasets. Notably, settings below 0.6 lead to performance declines in both cACC and sACC, likely due to the visual ambiguity given by the non-fine-tuned visual feature for the challenging fine-grained objects. For the Flower-102 dataset, a lower $\alpha$ improves clustering, in line with KMeans baseline results (Tab. 1). However, the sACC drop indicates a lack of semantic coherence in these clusters. Thus, we have chosen $\alpha = 0.7$ as the default setting for our system.

### G.2 ANALYSIS OF THE HYPERPARAMETER $K$

The hyperparameter $K$ dictates the number of augmented samples generated through a data augmentation pipeline featuring random cropping, color jittering, flipping, rotation, and perspective alteration. Our experiments across five fine-grained datasets, detailed in Fig. 17, reveal that additional augmented samples beyond a point fails to give further improvements. Moderate augmentation is sufficient to mitigate visual bias to a certain degree. Thus, we opt for $K = 10$ as our default setting to balance efficacy and computational cost.

### G.3 ANALYSIS OF VLM MODEL SIZE

In Fig. 18, we explore the relationship between the model size of the CLIP VLM utilized in FineR and our system performance. Our analysis reveals a direct, positive correlation between larger VLM

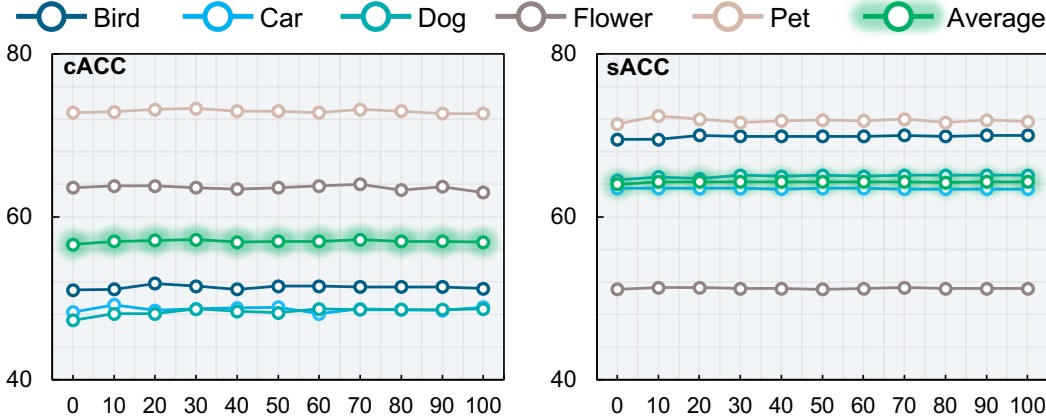

Figure 17: Sensitivity analysis of $K$ in FineR system. The average performance is highlighted in **Green** .

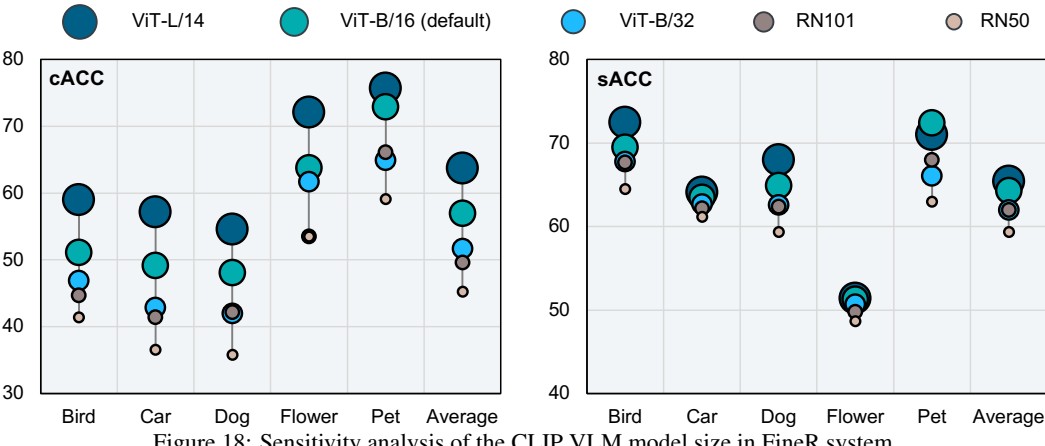

Figure 18: Sensitivity analysis of the CLIP VLM model size in FineR system.

models and enhanced system performance. Notably, when using the CLIP ViT-L/14 model, FineR registers an incremental performance gain of $+6.7\%$ in cACC and $+1.1\%$ in sACC, compared to the default ViT-B/16 model. This finding underscores the potential for further improvement in FineR through the adoption of more powerful VLM models.

### G.4  ANALYSIS OF THE NUMBER OF UNLABELED IMAGES USED FOR CLASS NAME DISCOVERY

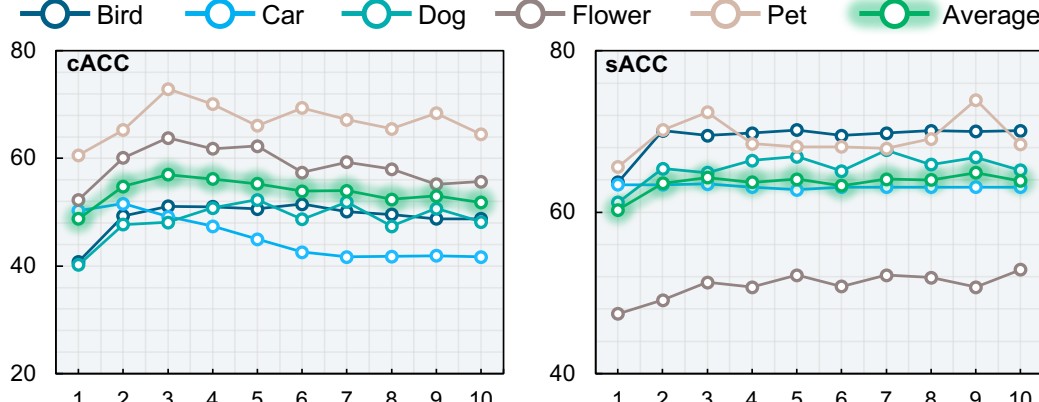

Figure 19: Sensitivity analysis of $\mathcal{D}_c^{\text{train}}$ in FineR system. The average performance is highlighted in **Green** .

In our fine-grained visual recognition (FGVR) framework, we operate under the practical constraint of having very limited unlabeled samples per category for class name discovery. Specifically,

$|\mathcal{D}^{\text{train}}|$ is much smaller than $|\mathcal{D}^{\text{test}}|$. While we explore both balanced ($|\mathcal{D}_c^{\text{train}}| = 3$) and imbalanced ($1 \leq |\mathcal{D}_c^{\text{train}}| \leq 10$) sample distributions in the main paper, an important question arises: does our FineR system require more samples for improved identification? We vary the number of unlabeled images per category, $|\mathcal{D}_c^{\text{train}}|$, from 1 to 10 and assessed system performance. As shown in Fig. 19, increasing the sample size does not lead to additional performance gains, indicating that FineR is efficient and performs well even with a sparse set of unlabeled images (*e.g.*, $|\mathcal{D}_c^{\text{train}}| = 3$).

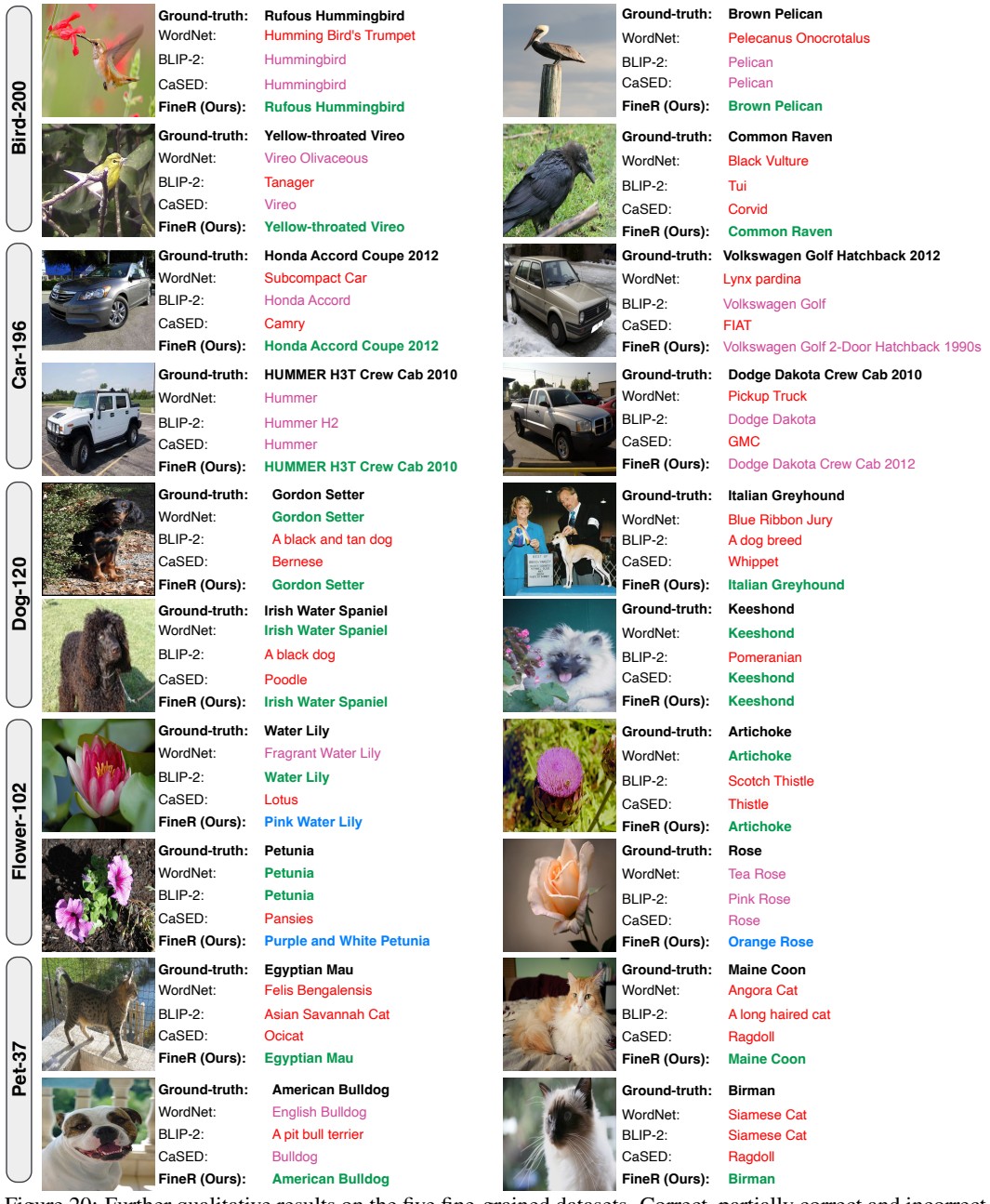

Figure 20: Further qualitative results on the five fine-grained datasets. Correct, partially correct and incorrect predictions are colored Green, Pink, and Red, respectively. Blue highlights the prediction that is even *more precise* than ground-truth category names.

## H   FURTHER QUALITATIVE RESULTS

We present additional qualitative results for the five fine-grained datasets in Fig. 20. The Car-196 dataset is one of the most challenging dataset due to the need to infer the specific year of manufacture of the car in the image. We find that on the Car-196 dataset, when neither our method nor other baseline methods give the completely correct answer, the partially correct answer given by our method is the closest to the correct answer (*e.g.*, "Dodge Dakota Crew Cab 2012" v.s. the ground-truth label "Dodge Dakota Crew Cab 2010"). Interestingly, similar to the qualitative visualization shown in Fig. 6, our FineR system can often give more precise and descriptive predictions even than the ground-truth labels (*e.g.*, "Orange Rose" v.s. the ground-truth label "Rose"). Through this additional qualitative analysis, we further demonstrate that our FineR system can not only accurately reason fine-grained categories based on the visual cues extracted from the discovery images, but also include visual details during its knowledge-based reasoning process, resulting in more precise and descriptive fine-grained discovery.

## I   FURTHER FAILURE CASE RESULTS

We further present a qualitative analysis of failure cases in Figure Fig. 21, compared with state-of-the-art vision-language models. We observe that, while most models fail to accurately identify and predict the correct names, our method often provides the most fine-grained predictions. These are not only closer to the ground truth class name but also capture the attribute descriptions more accurately, suggesting that our method makes 'better' mistakes. More interestingly, on the proposed novel Pokemon dataset, almost all compared methods predict only the real-world counterparts (*e.g.*,

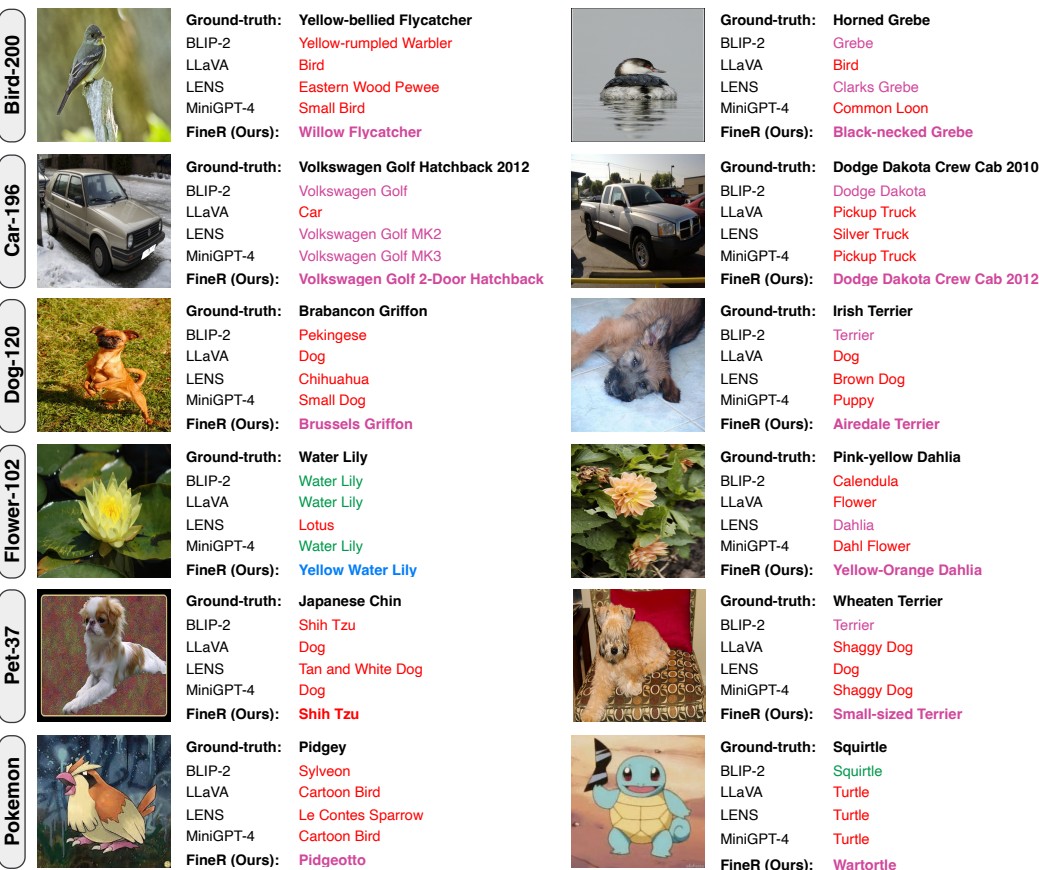

Figure 21: Further qualitative failure case results on the five fine-grained datasets and the Pokemon dataset. Correct, partially correct and incorrect predictions are colored Green, Pink, and Red, respectively. Blue highlights the prediction that is even *more precise* than ground-truth category names.

"Squirtle" v.s. "Turtle" or "Pidgey" v.s. "Bird")of the Pokemon characters, highlighting a domain gap between real-world and virtual categories. In contrast, although not completely accurate, our method offers predictions that are closer to the ground truth (evolutionized Pokemon characters) for the two Pokemon objects.

## J    HUMAN STUDY DETAILS

In this section, we describe the details of our human study, designed to establish a layperson-level human baseline for the FGVR task. Formally, we tailor two questionnaires in English for the Car-196 and Pet-37 dataset, respectively. In each questionnaire, we present participants with one image for each category within the target dataset. In the questionnaire, participants receive initial task instructions as follows: (i) Your task is to answer the name of the car (or pet) present in each image as specific as you can. You do not have to be sure about the specific name to answer; (ii) If you have no clue about the name of the car model (or pet breed) present in the image, you can describe and summarize how does this car (or pet) looks like in one sentence (*e.g.*, "It is a two-door black BMW sedan."). Recognizing that English may not be the first language of all participants, we include the prompt: "feel free to use your preferred translator for understanding or answering the question". Images paired with questions are then presented sequentially. To aid participants, we include the same attributes acquired by our FineR system beneath each question as a reference for describing the car (or pet). Two questions from the questionnaires are showcased in Fig. 22. We carry out this study with 30 participants. Given that the two questionnaires involve visual recognition and description of 196 and 37 images respectively, it takes each participant between 50 to 100 minutes to complete the study for both datasets. Leveraging the free-form text encoding abilities of CLIP VLM, we directly use the gathered textual responses to generate a text-based classifier for the two datasets from each participant's responses. We are thus able to perform evaluation inference on the two datasets using CLIP VLM along with the layperson classifiers. The layperson-level performance presented in Fig. 5 for both datasets are averaged across the 10 participants.

**Question 1:** What is the *model name* of the *car* in this photo (demo answer example: **BMW M2 CS 2020**)?

If you do not know the exact name, please describe the *visual "looks like"* of the car in the photo in one sentence. (demo answer: **This is a blue Porsche with two doors and a giant black tail.**)

**Visual attribute hints for better description:**
'approximate year of manufacture (e.g., 1990s, 2000s or 2010s)',
'possible make (automobile manufacturers)',
'doors', 'seats', 'windows',
'body style (Sedan, Wagon, SUV, Coupe, Roadster, Truck, Cab, Convertible, Minivan, Van, Hatchback, etc.)', 'body color', 'roof color', 'size', 'height', 'length', 'width',
'window size', 'window shape', 'window tint'
'emblem/logo on the front or the rear of the car', 'emblem/logo placement'
'grille design', 'grille shape', 'grille size', 'distinctive elements of the grille',
'headlight design', 'headlight shape', 'headlight size',
'taillight design', 'taillight shape', 'taillight size',
'wheel design', 'wheel size', 'wheel pattern',
'specific body panels, contours, or accent lines'
'roofline shape', 'door handle design', 'side mirror design', 'bumper design', 'hood design',

**Question 1:** What is the *breed name* of the *pet* in this photo? If you do not know the exact name, please describe the *visual "looks like"* of the pet in the photo in one sentence.

**Visual attribute hints for better description:**
'body shape', 'body size',
'head shape', 'head size', 'facial wrinkles',
'neck length',
'leg length', 'leg shape', 'leg thickness',
'paw size', 'paw shape', 'toe shape',
'tail shape', 'tail length', 'tail position',
'coat color', 'coat length', 'coat texture', 'coat pattern',
'fur texture', 'fur length' 'fur density',
'eye shape', 'eye color',
'ear position', 'ear shape',
'muzzle shape', 'snout length', 'nose shape', 'nose color',
'overall body shape and proportion',
'breed-specific markings or patterns (e.g. spots, stripes, patches)',
'overall appearance (e.g. sleek, fluffy, muscular)',
'facial markings', 'body markings',
'whisker length', 'whisker shape', 'claw length', 'claw shape'

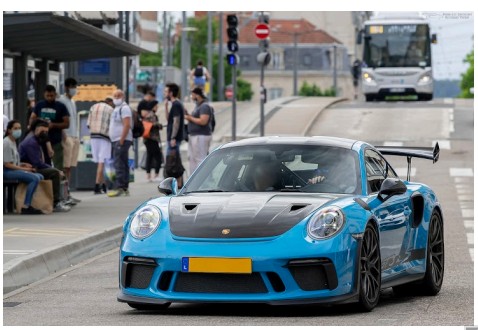
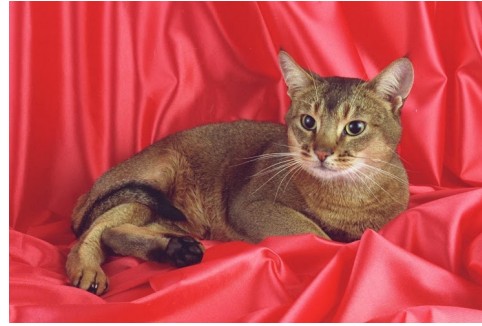

Car-196 Questionnaire Question Example         Pet-37 Questionnaire Question Example

Figure 22: Example questions from the human study questionnaires for the Car-196 and Pet-37 datasets.

# K FURTHER DISCUSSIONS

## K.1 REASONING CAPABILITY OF LARGE LANGUAGE MODELS

In a world awash with data, Large Language Models (LLMs) such as ChatGPT (OpenAI, 2022) and LLaMA (Touvron et al., 2023) have emerged as prodigious intellects of reasoning and problem solving capabilities (Brown et al., 2020; Wei et al., 2022; Kojima et al., 2022; Yao et al., 2023). They power transformative advances in tasks as diverse as common sense reasoning (Bian et al., 2023), visual question answering (Shao et al., 2023; Zhu et al., 2023a; Berrios et al., 2023), robotic manipulation (Huang et al., 2023), and even arithmetic or symbolic reasoning (Wei et al., 2022). *What if we could elevate these language-savvy powerhouses beyond the limitations of text and bestow upon them the 'gift of sight,' specifically for Fine-grained Visual Recognition (FGVR)?* In this work, we propose FineR system that translates useful visual cues for recognizing fine-grained objects from images into a language these LLMs can understand, and thereby unleashing the reasoning prowess of LLMs onto FGVR task.

## K.2 THE STORY OF BLACKBERRY LILY

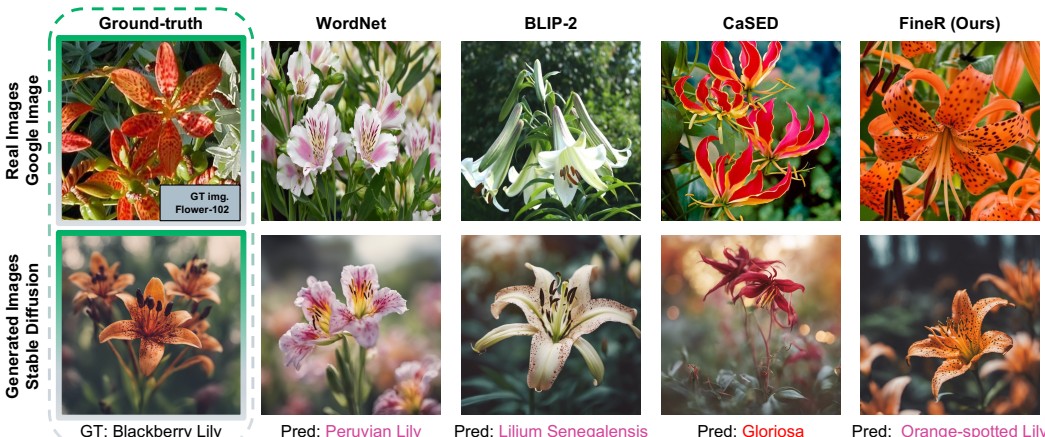

Figure 23: Reverse comparison of prediction results for the `"Blackberry Lily"` image (upper-left corner) in Flower-102. We evaluate the visual counterparts associated with the predicted semantic concepts. To conduct this comparison, we employ two distinct methods for inversely identifying their visual counterparts: (i) Google Image Search: we query and fetch images that are paired with the predicted class names from Google; (ii) Stable Diffusion: we utilize the predicted semantic class names as text prompts to generate semantically-conditioned images using Stable Diffusion. Partially correct and wrong predictions are color coded. None of the methods correctly predict the ground-truth label.

Predicted category names encapsulate valuable semantic information that should align with or align closely the ground truth. The predicted semantics will be used in downstream tasks. Therefore, a semantically robust prediction is important, even when it is a wrong prediction. For instance, a prediction of `"Peruvian Lily"` against the ground truth `"Blackberry Lily"` is more tolerable than incorrectly predicting `"Rose"`. As discussed in Sec. 3.1 , our FineR system showcases its capability for semantic awareness, particularly when all the methods wrongly predict a `"Blackberry Lily"`, it offers the most plausible prediction of `"Orange-spotted Lily"`. Nevertheless, a sole focus on textual semantics might not fully assess the quality of a prediction, which is important for downstream applications like text-to-image generation. We address this by additionally employing a unique methodology of reverse visual comparison using the prediction results of `"Blackberry Lily"`, as illustrated in Fig. 23. Specifically, we employ two techniques to locate visual analogs for the predicted classes: Google Image Search and Stable Diffusion (Rombach et al., 2022) text-to-image generation. The upper row in Fig. 23 shows images retrieved via Google Image Search using the predicted classes, while the lower row displays images generated by Stable Diffusion, conditioned on the predictions as text prompt.

This reverse comparison reveals that the visual counterparts retrieved through FineR are strikingly similar to the ground-truth, corroborating its robustness. In contrast, reversed prediction results from WordNet and BLIP-2 exhibit only partial similarities in petal patterns, aligning with their semi-accurate class names. The mispredictions from CaSED, however, lack such visual congruence. This simple yet insightful reverse comparison distinctly highlights the advantages of our reasoning-based

approach over traditional knowledge base methods. Accordingly, we proceed to discuss these two methodological paradigms as following.

**Knowledge Base Retrieval Methods: Forced Misprediction.** Knowledge base methods such as the WordNet baseline and CaSED rely heavily on the CLIP latent space to retrieve class names that are proximal to input images. They assume that the knowledge bases covers the knowledge (e.g., class names) for the target task. Despite the certain visual resemblance in Fig. 23, these methods overlook one of the most obvious visual attributes, such as `"color"`, in the input image. Thus, the reverse visual results are significantly different from the ground-truth. This underscores how the retrieval process is disproportionately influenced by the CLIP similarity score. If the CLIP latent space lacks a clear decision boundary for the given input, these methods are compelled to choose an incorrect class name from its knowledge base. This process is uninterpretable and ignores high-level semantic information. Consequently, such wrong predictions contain little useful visual information, leading to more egregious errors that are difficult to justify.

**Our Reasoning-based Approach: Making Useful Mistakes.** In contrast to knowledge-base methods, our FineR approach demonstrates semantic awareness, interpretability, and informativeness even in the face of errors. FineR system conditions its predictions on observed visual cues from the images, enriching the reasoning processes using LLMs. For example, if the visual cue `"color: orange"` is present, the model's reasoning avoids class names lacking this attribute. Even when FineR makes an incorrect prediction, it still provides semantically rich insights, such as the attribute `"Orange-spotted"`. Therefore, as shown in Fig. 23, our method not only offers textually informative results about the ground truth but also produces highly similar vision analogous, whether retrieved from Google Images or generated by Stable Diffusion. This confirms that FineR delivers high-quality, interpretable, and semantically informative predictions for both visual and textual domains, thereby enhancing the robustness of downstream applications.

### K.3    WORKING IN THE WILD

In this section we expand upon the comparison of our FineR with the LLM enchanced visual recognition methods such as PICa (Yang et al., 2022) or LENS Berrios et al. (2023). As mentioned earlier in Sec. 4, such systems first extract captions from an image using off-the-shelf captioning model and then feeds the question, caption and in-context examples to induce LLMs to arrive at the correct answer. In particular, LENS, in addition to captions, also builds visual vocabularies by collecting tags (or class names) from image classification, object detection, semantic segmentation and the visual genome datasets (Krishna et al., 2017). Examples of such datasets include ImageNet (Russakovsky et al., 2015), Cats and Dogs (Parkhi et al., 2012), Food 101 (Bossard et al., 2014), Cars-196 (Krause et al., 2013) and so on. Furthermore, LENS uses Flan-T5 (Longpre et al., 2023) to extract comprehensive descriptions from these class names, as proposed in the work of (Menon & Vondrick, 2023). Based on the vocabulary and image descriptions it uses CLIP to first tag a test image. Subsequently, it feeds the tags and the previously extracted captions to a LLM to arrive at an answer given a question. Aside from being unfair to FineR, where we do not assume a priori knowledge of any vocabulary, LENS has two major drawbacks: (i) Such an approach can only work if an image belongs to the well known concepts defined in a finite vocabulary, and will fail if the concept depicted in the image falls outside the vocabulary. (ii) Since CLIP is queried during the image tagging with such a large vocabulary, it shares the same disadvantages as the WordNet baseline, as the semantic vocabulary is quite unconstrained. To demonstrate this behaviour we show qualitative results in Fig. 24 where we compare LENS with FineR on five academic benchmarks and our Pokemon dataset. As hypothesized, LENS is really competitive to FineR on Bird-200, Car-196, Dog-120, Flower-102 and Pet-37 datasets, hitting the right answer several times as our FineR. This is due to the fact that LENS's vocabulary already contains the concepts present in the fine-grained datasets that are being tested for. Therefore, CLIP has no difficulty in zeroing in the right tag, followed by the right answer from the LLM. Such a trend has already been demonstrated with our UB baseline in Sec. 3.1. However, LENS fails miserably in the Pokemon dataset, misclassifying all of the six instances. For instance, most of the time the characters are classified as either real world animals (*e.g.*, "Bird" or "Newt") or as generic name "Pokemon". Conversely, our FineR do not face difficulty when presented with novel fine-grained concepts. Thus, unlike LENS, our FineR is designed to work not only in the academic setup but also in the wild. We believe that this versatility in FineR is indeed a big first step in democratizing FGVR.

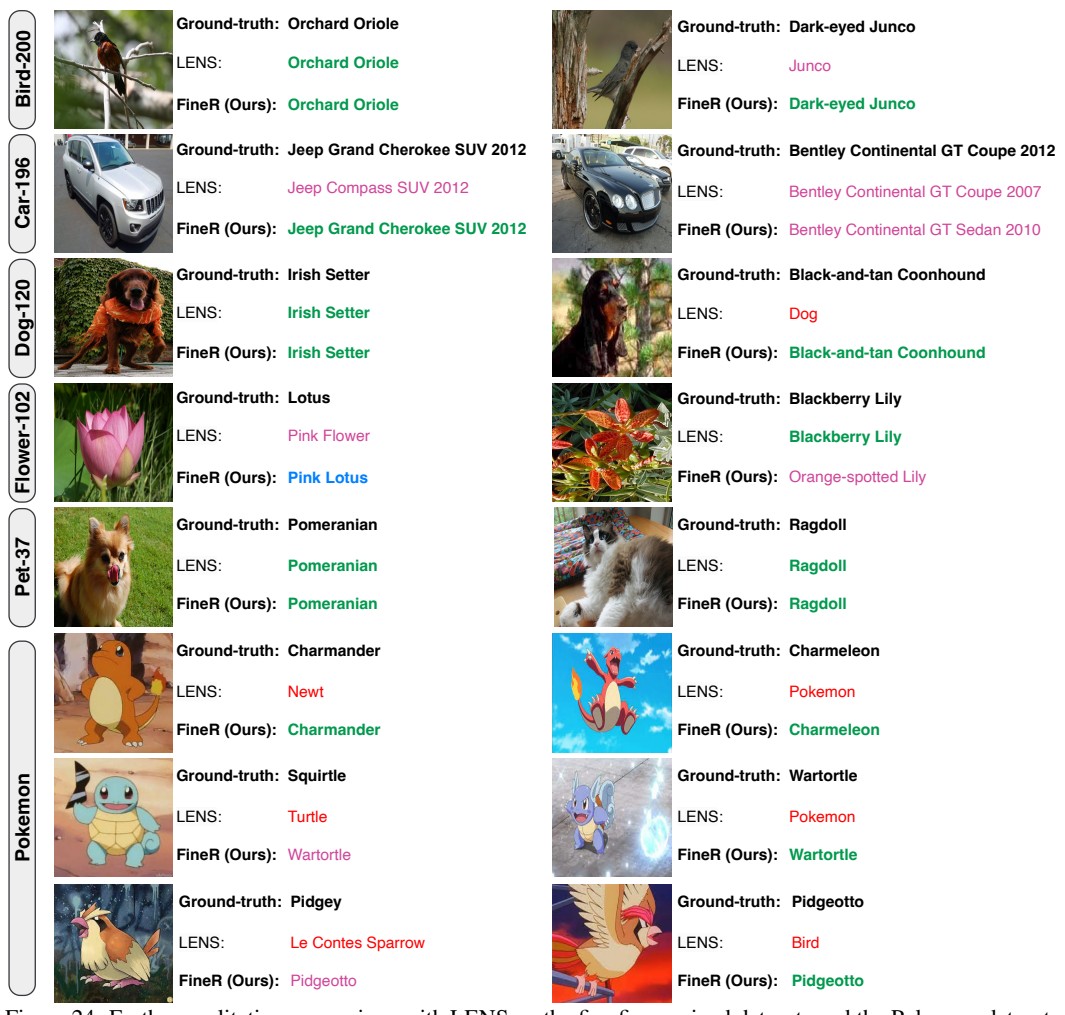

Figure 24: Further qualitative comparison with LENS on the five fine-grained datasets and the Pokemon datasets. Correct, partially correct and incorrect predictions are colored Green, Pink, and Red, respectively. Blue highlights the prediction that is even *more precise* than ground-truth category names.

## K.4 SETTING COMPARISON

**Zero-Shot Transfer and Prompt Engineering in Vision-Language Models.** The advent of vision-language models (VLMs) like CLIP (Radford et al., 2021) and ALIGN (Jia et al., 2021) has led to notable successes in zero-shot transfer across diverse tasks. To enhance VLM performance, the field of vision and learning has increasingly focused on prompt engineering techniques, encompassing prompt enrichment and tuning. Prompt enrichment methods, (Menon & Vondrick, 2023) and (Yan et al., 2023), leverages ground-truth class names to obtain attribute descriptions via GPT-3 queries. These descriptions are then utilized to augment text-based classifiers. Furthermore, prompt tuning methods, such as CoOp (Zhou et al., 2022b), TransHP (Wang et al., 2023), and CoCoOp (Zhou et al., 2022a), involve the automatic learning of text prompt tokens in few-shot scenarios with labeled data, achieving substantial enhancements over standard VLM zero-shot transfer methods. In contrast, our method only leverages *unlabeled* data in few-shot scenarios for discovering the class names, as the compared in Tab. 8. In addition, most importantly, as indicated in Tab. 9, all VLM-based methods and tasks discussed thus far necessitate *pre-defined class names as vocabulary* to work at inference time for constructing text classifiers. While such pre-defined vocabularies are less resource-intensive than expert annotations, they still require expert knowledge for formulation. For example, accurately defining 200 fine-grained bird species names for the CUB-200 dataset (Wah et al., 2011) is beyond the expertise of a layperson, thereby constraining the practicality and generalizability of these methods. In stark contrast, as compared in Tab. 9, our approach is **vocabulary-free**, operating *without* the need for pre-defined class name vocabularies. This absence of a pre-existing vocabulary renders traditional VLM-based methods *ineffective* in our challenging task. Instead, a *vocabulary-free* FGVR model in

| Setting | Model Training | Annotation Supervision | Training Data | |
|---|---|---|---|---|
| | | | Base Classes | Novel Classes |
| Standard FGVR | ✓ | Class Names | Full Labeled | ✗ |
| Generalized Zero-Shot Transfer | ✓ | Class Names + Attributes | Full Labeled | ✗ |
| VLM Zero-Shot Transfer | ✗ | ✗ | ✗ | ✗ |
| VLM Prompt Enrichment | ✗ | ✗ | ✗ | ✗ |
| VLM Prompt Tuning | ✓ | Class Names | Few Shots Labeled | ✗ |
| **Vocabulary-free FGVR (Ours)** | ✗ | ✗ | ✗ | Few Shots Unlabeled (*e.g.*, 3) |

Table 8: **Training setting comparison** between our novel vocabulary-free FGVR task and other related machine learning settings.

| Setting | Base Classes | Novel Classes | Output | Prior Knowledge |
|---|---|---|---|---|
| Standard FGVR | Classify | None | Names | Class Names |
| Generalized Zero-Shot Transfer | Classify | Classify | Names or Cluster index | Class Names and Attributes |
| VLM Zero-Shot Transfer | None | Classify | Names | Class Names |
| VLM Prompt Enrichment | None | Classify | Names | Class Names |
| VLM Prompt Tuning | Classify | Classify | Names | Class Names |
| **Vocabulary-free FGVR (Ours)** | None | Classify | Names | None |

Table 9: **Inference setting comparison** between our novel vocabulary-free FGVR task and other related machine learning settings.

our framework is tasked with automatically discovering class names from few unlabeled images (*e.g.*, 3-shots). Besides, our method does not need any prompt enrichment augmentation.

**Generalized Zero-Shot Learning (GZSL).** Under GZSL (Liu et al., 2018; 2023b) setting, a model is given a labeled training dataset as well as an unlabeled dataset. The labeled dataset contains instances that belong to a set of *seen classes* (as known as *base classes*), while instances in the unlabeled dataset belong to both the seen classes as well as to an unknown number of *unseen classes* (as known as *novel classes*). Under this setting, the model needs to either classify instances into one of the previously *seen classes*, or the *unseen classes* (or unseen class clusters Liu et al. (2018)). In addition, GZSL imposes additional assumption about the availability of prior knowledge given as auxiliary *ground-truth attributes* that uniquely describe each individual class in both base and novel classes. This restrictive assumption severely limits the application of GZSL methods in practice. Furthermore, the limitation becomes even more strict in fine-grained recognition tasks because attributes annotation requires expert knowledge. In contrast, as compared in Tab. 8, our novel FGVR setting approach is **training-free**, and does not require class names and attributes supervision from base classes and data. Most significantly, as highlighted in Tab. 9, our approach is distinctively **vocabulary-free**, thus eliminating the strong assumption of having access to class names and their corresponding attribute descriptions, which typically require specialized expert knowledge.

