# OpenReview forum: "Democratizing Fine-grained Visual Recognition with Large Language Models"
_ICLR.cc/2024/Conference — ICLR 2024 poster_

### Official Review · Reviewer_WYuk · 2023-10-18

**Soundness:** 3 good
**Presentation:** 3 good
**Contribution:** 2 fair
**Rating:** 6
**Confidence:** 5

**Summary:**

- This paper proposes a novel training-free FGVC pipeline.

- The key idea is to use multiple LLMs to generate the visual concepts to represent the imae of a fine-grained category, which allows not only trainin-free but also zero-shot inference.

- Besides, a novel Pokemon dataset is proposed to further foster FGVC.

- Extensive experiments on multiple standard FGVC datasets and the proposed dataset show the effectiveness of the proposed method.

**Strengths:**

- This paper is well-written, easy-to-follow and well-presented.

- A novel Pokemon dataset is proposed to benefit FGVC.

- The proposed method is novel and interesting, which provides a new way to do training-free FGVC based on LLM.

- The proposed method is experimentally better than existing state-of-the-art methods.

**Weaknesses:**

- Although the novelty and technique score is satisfactory for ICLR, a major issue is whether it is necessary to only leverage LLM for FGVC under the proposed setting. In fact, as far as the reviewer concerns, some other only learning prompt on VLM can already achieves more than 90% accuracy for zero-shot FGVC. For example:

[1] Conditional Prompt Learning for Vision-Language Models. CVPR 2022.

[2] Learning to Prompt for Vision-Language Models. IJCV 2022.

In fact, this issue is critical, at least from the vision community. What should be the correct role of LLM for FGVC? Should only the concepts from LLM be used for FGVC? Or it should be jointly used with visual representation to aid FGVC? Soley relying on LLM for vision tasks is in a way that the reviewer does not appreciate.

- The idea to use LLM to generate concepts for FGVC, is already not new now. Some recent works implement idea:

[3] Learning concise and descriptive attributes for visual recognition. ICCV 2023.

- The proposed dataset also has multiple issues for publication. For example:

(1) The exact number of samples, how the training and testing set is divided, and etc. Many details are missing in the main submission and the supplementary materials.

(2) As the dataset is Pokemon, not real-world FGVC categories, some further questions raise: Is it able to describle the fine-grained patterns? Besides, is the LLM leart from real-world able to differentiate the fine-grained patterns on Pokemon. These issues make the contribution of FGVC to be doubted.

**Questions:**

My concerns on Q1, 2, 3 have been well addressed. Although I still have some minor concerns, it does not impede that this work has reached the accept threshold. So, I decide to improve my rating to borderline accept.

%%%%%%% before rebuttal

Although the strength seems to overwhlem the weakness, as an expert in vision and FGVC, the reviewer still holds a critical view towards this LLM based work, which may be good enough for pulibcation. Some specific questions:

Q1: As there is already some good VLM solution to do zero-shot FGVC and achives nearly 90% accuracy, is it necessary to solely address FGVC by pure concepts from LLM? Besides, is it necessary to solely rely on LLM instead of VLM or visual representation for FGVC?

Q2: The idea of this paper is actually not very new now. The ICCV paper [3] implements similary idea. The difference and comparison should be made and clarified.

Q3: So many details of the proposed dataset are missing. The specific comments have been listed in the weakness.

Q4: Is the proposed Pokemon dataset can really fit the fine-grained patterns? Or, the LLM and VLM learnt from real-world data, can really discriminate the fine-grained pokemon? These issues may make its contribution actually limited for FGVC community.

---

> ### Author Response · Authors · 2023-11-18
> **Response to Reviewer WYuk (1/4)**
>
> Thank you for your positive, attentive and constructive feedbacks! We are encouraged that you, as an expert in vision and FGVC, recognized our method as "**novel and interesting**" and "experimentally better than existing state-of-the-art methods" which provides "a new way to do training-free FGVC based on LLM". We address your thoughts point by point below.
>
> >**Q1:** As there is already some good VLM solution to do zero-shot FGVC and achives nearly 90% accuracy, is it necessary to solely address FGVC by pure concepts from LLM? Besides, is it necessary to solely rely on LLM instead of VLM or visual representation for FGVC?
> >
> >**A1:** Great question! Thanks for recoginizing the novelty and technique score of our work are statisfactory for ICLR. Following your suggestion, we have now also added a discussion section in **Appendix K.4** of the revision to carefully compare our setting and method with respect to VLM zero-shot transfer and prompt enrichment/tuning methods. Here, we breakdown this question into three subsequent questions and address them below:
> >
> >1. **Setting perspective:** In fact, as recoginized by ```Reviewer oRxX```, we would like to emphasize and clarify that the novel task we propose and study in this work is **vocabulary-free FGVC with only few unlabeled samples as natural observation**. We compare our setting and existing VLM solutions apple-to-apple in the **Table 1 and Table 2 (in Response 2/4)**. To be more specific, under vocabulary-free FGVC setting, there is no target class names (vocabulary) pre-defined by domain experts (*e.g.*, ornithologist) for VLM-based zero-shot transfer methods or VLM prompt tuning methods to construct classifiers. This makes existing VLM solutions (*e.g.,* CLIP, CoOp[2], CoCoOp[1]) that rely on the pre-defined vocabulary **fail to function** in this novel task, although VLM solutions such as [1] and [2] can achieves good performance for standard zero-shot FGVC task where the class names are already known. To be best of our knowledge, vocabulary-free FGVC task is underexplored in the existing literature. We believe this is an important yet realistic novel task for the community to explore. This newly proposed vocabulary-free FGVC task is inspired by the natural case:
> >
> > > "An bird lover gathered several unlabeled images from a webcam set up in the Amazon jungle. Confronted with the task of identifying the fine-grained bird species existing the the region from these few unlabeled observations, and without any pre-defined species names as prior knowledge from ornithologists, the bird lover encountered a significant challenge. In this context, our proposed FineR system is functional, enabling the bird lover to not only discover the bird species but also to classify them effectively in the incoming data stream from the webcam.“
> >
> >2. **Method perspective:** Our proposed method **is not** solely rely on LLM. In fact, the proposed FineR system jointly integrates the visual information translation capability of the **VQA model**, the reasoning capability of the **LLM**, and the zero-shot discrimination capability of **contrastive VLM** to tackle the proposed vocabulary-free FGVR task. To further clarify, FineR primarily operates in 3 steps: (i) first uses BLIP-2 to **translate** visual information (super-category and attribute descriptions) from the few (3 per class) unlabeled images to natural language; (ii) then uses the visual cues in text to elicit the reasoning capability of ChatGPT for **discovering fine-grained categories**; (iii) lastly uses CLIP to classify/recognize fine-grained categories in the test data stream based on the discovered concetps.
> >
> >3. **The necessity of using LLMs:** Insightful question! In our work, **the role of LLM is a Reasoning Engine**, just like a human expert would reason fine-grained categories based on the visual input. Furthermore, to further study "Is it necessary to use LLMs?", we constructed strong baseslines that adopt different solutions: (i) Learning-based Sinkhorn; (ii) Knowledge base based WordNet (use CLIP to retrive categories from WordNet); (iii) and BLIP-2 (VQA), as well as compared  with SOTA methods (SCD and CASED) based on large corpus knowledge bases. As shown in the quantitative results in Tab. 1 and qualitative results in Fig. 6 of the paper, our FineR system not only performs better but also makes more plausible mistakes. This demonstrates the advantages of using LLMs to **reason** in vocabulary-free FGVC task.
>
> [1] Conditional Prompt Learning for Vision-Language Models. CVPR 2022.
> [2] Learning to Prompt for Vision-Language Models. IJCV 2022.

---

> ### Author Response · Authors · 2023-11-18
> **Response to Reviewer WYuk (2/4)**
>
> **Table 1: *Training* setting comparison between our novel vocabulary-free FGVR task and other related machine learning settings.**
> |                                     | **Model**               | **Annotation**            | Data    | Data       |
> | ----------------------------------- |:------------------:| :--------------: | :------: | :-------------: |
> | **Settings**                        | **Training** | **Supervision**  | **Base Classes** | **Novel Classes** |
> | Standard FGVR Classification                    | &#10004;  |  Class Names + Auxilliary attributes    | Fully Labeled    | &#10006;     |
> | VLM Zero-shot transfer              | &#10006;  | &#10006;     | &#10006;     | &#10006;     |
> | VLM Prompt Enrichment [3]           | &#10006;  | &#10006;     | &#10006;     | &#10006;     |
> | VLM Prompt Tuning ([1], [2])        | &#10004;  | Class Names   | Few (e.g., 16) Shots Labeled    | &#10006;     |
> | **Vocabulary-free FGVR (Ours)**     | &#10006;  | &#10006;       | &#10006;     | **Few (e.g., 3) Shots unlabeled** |
>
> **Table 2: *Inference* setting comparison between our novel vocabulary-free FGVR task and other related machine learning settings.**
>
> | Setting                         | Base Classes     | Novel Classes        | Output                         | Prior Knowledge                |
> | ------------------------------- | ---------------- | -------------------- | ------------------------------ |------------------------------- |
> | Standard FGVR Classification                    | &#10004;  |  Class Names    | Full Labeled    | &#10006;     |
> | VLM Zero-Shot Transfer          | None             | Classify             | Names                          | Class Names                    |
> | VLM Prompt Enrichment [3]       | None             | Classify             | Names                          | Class Names                    |
> | VLM Prompt Tuning ([1], [2])    | Classify         | Classify             | Names                          | Class Names                    |
> | **Vocabulary-free FGVC (Ours)** | **None**         | **Classify**         | **Names**                      | **None**                       |

---

> ### Author Response · Authors · 2023-11-18
> **Response to Reviewer WYuk (3/4)**
>
> >-----
> >**Q2:** The idea of this paper is actually not very new now. The ICCV paper [3] implements similary idea. The difference and comparison should be made and clarified.
> >
> >**A2:** We would like to clarify that the settings and methods of our paper and [3] are very different.
> > * **Setting**. The work in [3] assumes the fine-grained categories are **known**. On the contrary, we do not assume any knowledge about the class names.
> > * **Implementation**. In [3], a LLM is queried to simply enhance the descriptions of the already known concepts (or class names) to improve classification. Instead, we use VQA model to first identify super-category (e.g., "Bird") from an image and then acquire useful attribute names (no description) for distinguishing the fine-grained sub-categories. Then we use a VQA model to describe the attribute values in the image. The acquired attribute-description paris are then used to invoke LLM reasoning for discovering the fine-grained categories that are **unknown**. **In a nutshell, [3] only uses LLM to describes known classes, while our method uses the interaction between LLM and VLM to reason unknown classes.**
> >
> >In fact, [3] falls into the same group of LLM-based VLM prompt enrichment methods as the ICLR 2023 work [4] that we discussed in the second paragraph of Section 4. We have now also included [3] in the second paragraph Line#3 of Section 4 in the revision. Thanks for the suggestion!
>
> [3] Learning concise and descriptive attributes for visual recognition. ICCV 2023.
> [4] Visual Classification via Description from Large Language Models. ICLR 2023.
>
> >-----
>
> > **Q3:** So many details of the proposed dataset are missing. The specific comments have been listed in the weakness.
> >
> > **A3:** In fact, **we have included the exact number of samples, train/test splits, and statistics of the proposed Pokemon dataset in Appendix A.2**. Furthermore, due to the small scale of the proposed Pokemon dataset, which consists of 10 categories with 3 images each for discovery (D_train) and 10 images per category for testing (D_test), we **direcly present the entire dataset and its splits in Appendix A.2, Fig. 9**. It is important to note that the Pokemon dataset was designed to delve deeper into the capabilities of vocabulary-free FGVC in the wild. Despite its small scale, it poses significant challenges for many strong methods we compared. We will elaborate on these challenges in the next question.

---

> ### Author Response · Authors · 2023-11-18
> **Response to Reviewer WYuk (4/4)**
>
> >**Q4:** Is the proposed Pokemon dataset can really fit the fine-grained patterns? Or, the LLM and VLM learnt from real-world data, can really discriminate the fine-grained pokemon? These issues may make its contribution actually limited for FGVC community.
> >
> >**A4:** Excellent question! The second part of your question precisely captures the motivation behind creating this dataset. Our intent was exactly to further examine the ability of our method, as well as the compared methods, to discover and differentiate virtual and wild FGVC categories effectively. Therefore, we proposed this wild and virtual Pokemon dataset, which is relatively novel and challenging for foundation models. We breakdown this question into two subsequent questions and address them below:
> >* **Fine-grained Patterns:** We mannuly selected 10 highly visually similar Pokemon categories to capture the fine-grained patterns. As shown in Appendix A.2, Fig. 9, since two Pokemon categories evolved from each other, their inter-class variance is low (*e.g.*, the subtle differences between "Pidgey" and "Pidgeotto" are only at the crown and tail.) In addition, since Pokemons are cartoon characters, they by default have high inner-class variance (*e.g.*, different facial expressions or body decorations). Additionally, the proposed Pokemon dataset presents an extra intriguing challenge due to the close resemblance of Pokemon characters to their real-world animal counterparts (e.g., "Squirtle" closely resembles a "Turtle" and "Pidgey" a "Bird"). This cross-domain similarity introduces an additional domain gap challenge in FGVC discovery. As illustrated in Fig.7, methods based on knowledge bases often incorrectly identify the real-world counterpart of the Pokemon (e.g., identified 'Turtle' instead of 'Squirtle'). In stark contrast, our FineR system successfully discovered 7/10 ground-truth Pokemon categories.
> >
> >* **the LLM and VLM learnt from real-world data, can really discriminate the fine-grained Pokemons?**: Yes! As demonstrated in Fig.7, our FineR system successfully reasoned and discovered 7/10 ground-truth and subsequently achieved upper-bound clustering accuracy and high semantic similarity accuracy. This suggests that LLMs and VLMs trained with Internet-scale data are able to discriminate the fine-grained virtual concepts such as Pokemons. We found the results of this experiment fascinating!
>
> We hope our answers have adequately addressed your concerns. We hope you might consider raising your score for our pioneering work in the vocabulary-free FGVC task. Should you have any additional questions, please don't hesitate to let use know. We would be more than happy to engage in further discussion with you. Thanks again for your insightful feedbacks strengthening our paper and stimulating our thinking.

---

> > ### Comment · Reviewer_WYuk · 2023-11-18
> > **Response to Authors  from Reviewer WYuk**
> >
> > Thanks for the authors for providing such a detailed rebuttal.
> >
> > Yes, my concerns on: 1) its necessaity when compared with existing zero-shot inference by VLM such as CLIP, BLIP; 2) its difference and insights against the recent ICCV 2023 paper have been well clarified.
> > So, after the clarification, I think this work has already meet the accept threshold.
> >
> > Although I am still not fully convinced that the Pokemon dataset may have a significant contribution to real FGVC tasks, and think that leveraging the concepts from LLM to improve the visual representation may be more pratical for FGVC, these concerns (more personal) do not necessarily impede the objective rating for the work itself.
> >
> > Thus, I would like to improve my rating to above the accept threshold.

---

> > > ### Author Response · Authors · 2023-11-18
> > > **Thanks for your acknowledgment**
> > >
> > > Dear Reviewer WYuk,
> > >
> > > We greatly appreciate your helpful comments and your satisfaction with our responses, particularly regarding the comparison with existing settings! We will add the above important discussions in the final manuscript and highlight them. We really enjoy communicating with you and appreciate your efforts.

---

### Official Review · Reviewer_3hkC · 2023-10-24

**Soundness:** 3 good
**Presentation:** 3 good
**Contribution:** 2 fair
**Rating:** 6
**Confidence:** 4

**Summary:**

This paper proposes a fine-grained semantic category reasoning method for fine-grained visual recognition, which attempts to leverage the knowledge of large language models (LLMs) to reason about fine-grained category names. It alleviates the need of high-quality paried expert annotations in fine-grained recognition.

**Strengths:**

1.	The presentation of this paper is clear and the proposed method is easy to follow.
2.	This paper proposes to extract visual attributes from images into the large language models (LLM) for reasoning the fine-grained subcategory names, which is a promising way to alleviate the high need for expert annotations in fine-grained recognition.

**Weaknesses:**

The weaknesses are as follows：

1.	The novelty of this paper should be further demonstrated. The proposed method seems an intuitive combination of existing large-scale models, such as the visual question answering model, large-language model and vision-language model, etc. Besides, extracting visual attributes from images for recognition is widely used in generalized zero-shot learning methods such as [a].

2.	The effectiveness of the proposed method should be further verified. The recognition accuracy on the CUB dataset is relatively low. Besides, existing generalized zero-shot learning methods such as [a] should be compared and more corresponding analyses should be added.

3.	There lacks a complete recognition example for explaining the whole procedure, which should be added for better understanding.

[a] Progressive Semantic-Visual Mutual Adaption for Generalized Zero-Shot Learning. CVPR 2023

**Questions:**

The novelty should be clarified and the difference from existing generalized zero-shot learning methods using visual attributes for recognition should be explained. More corresponding experiments and analyses should be added.

---

> ### Author Response · Authors · 2023-11-18
> **Response to Reviewer 3hkC (1/3)**
>
> Thank you for your feedback. We are particularly encouraged with your comment about our method: "is a promising way to alleviate the high need for expert annotations in fine-grained recognition". We address your feedback point by point below.
>
> >**Q1.1:** The novelty should be clarified.
> >
> >**A1.1:** Thanks for the suggestions. Our work, we believe, is novel in several key aspects:
> >* **Innovative Application/Task:** To the best of our knowledge, our work is the first to tackle the novel task of **vocabulary-free FGVR with only a few unlabeled samples as natural observation**. To tackle this novel task we harness the reasoning capabilities of LLMs and leverage the Internet-scale expert knowledge encoded in their weights to first discover the fine-grained categories that are present in a dataset, which is otherwise not know a priori. While CaSED (Conti et al., 2023) previously addressed vocabulary-free classification by retrieving concepts from a knowledge-base, our method is tailored specifically for FGVR by leveraging the knowledge in LLM. This also resonates with the comments from `Reviewer 3hkC` and `Reviewer WYuk` who commented that utilization of LLM for FGVR as "**a new way**" and "**a promising way**", respectively.
> >
> >* **Methodological Advancements:** We introduced **FineR**, a novel system that integrates the visual information translation capability of VQA models, the reasoning capability of LLMs, and the zero-shot discrimination capability of contrastive VLMs to address the vocabulary-free FGVR task. This approach is recognized by `Reviewer WYuk` as "**novel**" and "**interesting**". Furthermore, as shown in the preliminary experiments in Fig. 2, we observed that other LLM-based methods requiring training alignments (such as BLIP-2 learning Q-former, LLaVA, and MiniGPT-4 learning a projector) struggle to discover fine-grained categories effectively due to the limitations of alignment data. These training-based methods cannot fully utilize the reasoning capabilities of LLMs. In contrast, our FineR system translates visual information from unlabeled images into natural language in order to fully harness the reasoning power of LLMs for fine-grained concepts. Thus, we showed that visual and textual concepts can be aligned *without* the need of expensive training.
> >
> >* **Resource Efficiency and Accessibility:** Our approach is both training-free and vocabulary-free, significantly reducing the computational and data resources typically required for vision model and LLM alignments. This enhances the accessibility and feasibility of our method for a broader range of users, potentially democratizing the application of FGVR across various sub-fields.
> >
> >* **Scalability and Generalization:** Being training-free enhances the scalability and generalizability of our methods. As demonstrated in Fig. 7 (Section 3.2, page 8) and Fig. 24 (Appendix K.3, page 31), our method can effectively discover wild and novel fine-grained Pokémon categories, outperforming other methods that fail in these wild scenario.
> >
> > > We have summarized our novelty and contributions at the end of the Introduction section (Sec. 1) in the revision.
> >-----

---

> ### Author Response · Authors · 2023-11-18
> **Response to Reviewer 3hkC (2/3)**
>
> >**Q1.2:** The difference from existing generalized zero-shot learning methods using visual attributes for recognition should be explained.
> >
> >**A1.2:** **In Table 1 and Table 2 below**, we have compared our vocabulary-free FGVR setting with Generalized Zero-Shot Learning (GZSL) at training and inference time, respectively. Furthermore, we have now also added a discussion section in **Appendix K.4** of the revision to carefully compare our setting and method with respect to GZSL.
> >
> **Table 1: *Training* setting comparison between our novel vocabulary-free FGVR task and other related machine learning settings.**
> |                                     | **Model**               | **Annotation**            | Data    | Data       |
> | ----------------------------------- |:------------------:| :--------------: | :------: | :-------------: |
> | **Settings**                        | **Training** | **Supervision**  | **Base Classes** | **Novel Classes** |
> | Generalized Zero-shot Learning ([a])      | &#10004;  | Class Names + Attributes    | Full Labeled    | &#10006;     |
> | **Vocabulary-free FGVR (Ours)**     | &#10006;  | &#10006;       | &#10006;     | **Few (e.g., 3) Shots unlabeled** |
> >
> **Table 2: *Inference* setting comparison between our novel vocabulary-free FGVR task and other related machine learning settings.**
>
> | Setting                         | Base Classes     | Novel Classes        | Output                         | Prior Knowledge                |
> | ------------------------------- | ---------------- | -------------------- | ------------------------------ |------------------------------- |
> | Generalized Zero-Shot Learning ([a])  | Classify         | Classify             | Names or Cluster Index (novel) | Class Names and Attributes     |
> | **Vocabulary-free FGVR (Ours)** | **None**         | **Classify**         | **Names**                      | **None**                       |
>
> > Generalized Zero-Shot Learning (GZSL) methods, like [a], rely on **pre-existing ground-truth attribute-description pairs** for both seen and unseen classes. GZSL methods use these attributes and seen (base) classes to train a vision model that can generalize to unseen classes based on shared attributes. Our method diverges significantly from GZSL in several ways:
> >* **Attribute and Description Availability**: As shown in **Table 1 and Table 2 above**, in our approach, **neither ground-truth attribute names nor descriptions are available**; thus, we do not use any predefined attribute information as prior knowledge, which requires expert annotations.
> > * **Attribute and Description Acquisition**: Initially, our method identifies super-category names (e.g., "Bird") using a VQA model, followed by querying an LLM to **generate relevant attribute names** for further distinguishing fine-grained sub-categories. **The attribute descriptions are then extracted from a few unlabeled images using the VQA model**.
> >* **Attribute and Description Utilization**: The generated **attribute-description pairs serve as high-level semantic cues in natural language**, invoking LLM reasoning in identifying fine-grained categories. This approach grants our method semantic awareness, enabling it to provide useful semantic information even in cases of incorrect predictions, as demonstrated in Fig.6.
> >-----

---

> ### Author Response · Authors · 2023-11-18
> **Response to Reviewer 3hkC (3/3)**
>
> >**Q2:** The effectiveness of the proposed method should be further verified. The recognition accuracy on the CUB dataset is relatively low. Besides, existing generalized zero-shot learning methods such as [a] should be compared and more corresponding analyses should be added. More corresponding experiments and analyses should be added.
> >
> >**A2:** We first would like to clarify that it is not fair to directly compare between our approach with generalized zero-shot learning methods. It's crucial to highlight that our approach is both **vocabulary-free** and **training-free**, which fundamentally differs from GZSL methods. Directly comparing our method's recognition accuracy with that of GZSL approaches, which are **supervised** on numerous base seen classes with class name and attribute annotation and jointly evaluated on both seen and unseen classes, would be inappropriate. For instance, GZSL methods' accuracy on the CUB-200 dataset is jointly evaluated on 150 seen and 50 unseen classes. In contrast, all 200 classes of CUB are unseen in our evaluation. Moreover, as outlined in the setting comparison in **Table 1 and Table 2 above**, all the prerequisites required by GZSL during training and inference, including class names, annotated base class data, and ground-truth attribute-description pairs are not available in our setting. In our vocabulary-free FGVR task, both GZSL and standard VLM zero-shot transfer methods, which depend on predefined vocabulary, **fail to function** due to the absence of class names. A thorough review of [a] leads us to conclude that a comparison with **GZSL approaches is not feasible in our setting due to the absence of class names**. In fact, standard VLM zero-shot transfer method (CLIP) was considered and present as the upper bound in our experiments. Notably, we have achieved a significant performance improvement on the CUB-200 dataset compared to the methods we evaluated against. Nevertheless, **we appreciate your suggestion and have added a discussion section in Appendix K.4** of the revised manuscript to specifically clarify the differences between our vocabulary-free FGVR setting and GZSL, along with comparisons to other related settings.
>
> >-----
>
> >**Q3:** There lacks a complete recognition example for explaining the whole procedure, which should be added for better understanding.
> >
> >**A3:** Thank you for your valuable suggestions! Indeed, **we have demonstrated the entire recognition process using an unlabeled "Italian Greyhound" image as an example in Appendix B.2, from Fig.12 to Fig.15**. This includes the visual input, the identified super-category ('Dog'), attribute names obtained from the LLM, attribute descriptions extracted by the VQA model, the complete LLM reasoning prompt, and the LLM's output results. In response to your advice, we have added a new figure in **Appendix B.2 Fig.11** of the revised manuscript to more clearly illustrate the full pipeline with actual inputs and outputs.
>
> We hope our responses have sufficiently addressed your concerns. With these clarifications and changes in the revision based on your suggestion, we kindly hope you might be convinced to raise your score for our pioneering work in the vocabulary-free FGVC task. Your attentive feedback has been helpful in enhancing our paper in the revision, and for that, we are truly grateful.
>
> [a] Progressive Semantic-Visual Mutual Adaption for Generalized Zero-Shot Learning. CVPR 2023.

---

> ### Author Response · Authors · 2023-11-21
> **Looking Forward to Your Reply**
>
> Dear Reviewer 3hkC,
>
> We genuinely appreciate the valuable feedback you have provided.
>
> As the deadline for the Author-Reviewer Discussion Phase approaches, we are reaching out to inquire if you might have any remaining questions or concerns regarding our submission.
>
> We wholeheartedly thank you for your dedication and effort in evaluating our submission. Please do not hesitate to let us know if you need any clarification or have additional suggestions. Your input is highly valued.
>
> Best regards,
>
> Authors of Submission #2720

---

### Official Review · Reviewer_oRxX · 2023-11-04

**Soundness:** 4 excellent
**Presentation:** 4 excellent
**Contribution:** 3 good
**Rating:** 8
**Confidence:** 5

**Summary:**

FineR leverages large language models to identify fine-grained image categories without expert annotations, by interpreting visual attributes as text. This allows it to reason about subtle differences between species or objects, outperforming current FGVR methods. It shows potential for real-world applications where expert data is scarce or hard to obtain.

**Strengths:**

This is a good paper, and the advantages are:
1. The utilization of LLM to vocabulary-free FGVR tasks is novel;
2. The paper is generally well-written and easy to follow;
3. Good performance in Table 1
4. The well utilization of LLM.

**Weaknesses:**

Advice:
1. Add the citation of some highly related missing works:
(1) Transhp: image classification with hierarchical prompting; it also focuses on the fine-grained image classification task. Also, it takes advantage of the recently proposed prompting technique in CV. Is your used LLM better or prompting better?
(2) V2L: Leveraging Vision and Vision-language Models into Large-scale Product Retrieval; it also focuses on the vision language model and fine-grained visual recognition. The paper is the champion of FGVC 9. A discussion of comparison seems necessary.
2. Could you show more examples of Fig. 2? Or any failure cases of Fig. 2?
3. What is your view of NOVELTY of a work using LLMs without training anything?

**Questions:**

I want to discuss with the authors that:
What is your view of NOVELTY of a work using LLMs without training anything?

---

> ### Author Response · Authors · 2023-11-18
> **Response to Reviewer oRxX (1/2)**
>
> We sincerely thank you for your positive feedback and helpful suggestions! We are glad that you recognize our method, which utilizes LLM for tackling **vocabulary-free FGVR tasks**, both "novel" and demonstrating "good performance", as well as representing a "well utilization of LLM". Furthermore, we appreciate that you found that our proposed method has *potential to work in real-world applications*. We address your questions below.
>
>
> >**Q1:** Add the citation of some highly related missing works.
> >
> >**A1:** Thanks you for your helpful advice. In the revision, we have now included these two highly related papers in the second paragraph of Related Work section (Sec.4) of the main paper. Furthermore, we have now also added a discussion section in Appendix K.4 of the revision to carefully compare our setting and method with respect to related works. In response to your question, we discuss the main differences between our work and TransHP and V2L below:
> >* **TransHP**: We note that the very recent NeurIPS'23 FGVR work TransHP (arXived on 2023-October-23, which is after ICLR'24 deadline) is related to our study. Our method differs from TransHP in two ways: (i) We study the task of **vocabulary-free** FGVR, where neither coarse-grained class names nor the target fine-grained class names are known, unlike TransHP where coarse grained classes are known; (ii) Our method is also **training-free**. The only resources our method has access are few unlabeled images (3 per-class or [1,10] per class) as an observation for discovering the fine-grained class names. Under this realistic vocabulary-free FGVR setting TransHP *cannot operate* due to the lack of class names.
> >
> >
> >* **V2L**: V2L learns and complementarily ensembles vision (ResNeSt, ResNeXt, CotNet, HS-ResNet, NAT, ViT, and ViT) and vision-language models (BLIP, ALBEF, XVLM, METER and SLIP) with the supervision from coarse-grained labels and textual description. Similar to TransHP, the lack of class names and supervisory signals makes V2L inoperable in our vocabulary-free scenario.
>
>
> >-----
>
> >**Q2:** Could you show more examples of Fig. 2? Or any failure cases of Fig. 2?
> >
> >**A2:** In the Appendix Fig. 20 we have provided more examples of Fig. 2. In Fig. 20 we have compared our method with several baselines. We observe that our FineR predicts the correct class more number of times than the baselines.
> >
> >In addition, following your suggestion we have added a thorough qualitative failure case analysis in **Appendix I, Fig. 21**. Specifically, Fig. 21 qualitatively presents the failure cases of our method across each dataset (including five fine-grained datasets and the proposed Pokémon dataset) and compares these with vision-language models (BLIP-2, LENS, LLaVA, MiniGPT-4). We observe that most of the baselines fail to correctly identify the finegrained categories. On the contrary our FineR, albeit not being able to predict the exact fine-grained category, it can still capture some fine-grained attributes. For eg., it predicts the dog breed *Brabancon Griffon* as *Brussels Griffon*, demonstrating that it can capture the characteristics of a *Griffon* dog. This suggests that our FineR makes *less severe* mistakes than the baselines.
> >
> >Intriguingly, on the novel Pokémon dataset, almost all compared methods predict only the real-world counterparts of the Pokémon characters (e.g., 'Squirtle' as 'Turtle' and 'Pidgey' as 'Bird'), highlighting a domain gap between real-world and virtual categories. In contrast, although not completely accurate, our method offers predictions that are closer to the ground truth (evolutionized Pokemon characters, 'Pidgey' as 'Pidgeotto' and 'Squirtle' as 'Wartortle') for the two Pokémon objects. This failure case analysis further confirms that our system effectively captures fine-grained visual details from images and leverages them for reasoning. It demonstrates that FineR not only generates more precise and fine-grained predictions when correct but also displays high semantic awareness when it errs.

---

> ### Author Response · Authors · 2023-11-18
> **Response to Reviewer oRxX (2/2)**
>
> >-----
> >**Q3:** What is your view of NOVELTY of a work using LLMs without training anything?
> >
> >**A3:** Thank you for your insightful question! Following your suggestion, we have outlined the novel aspects of our work at the end of the Introduction section (Sec. 1) in the revision. Our work, we believe, is novel in several key aspects:
> >* **Innovative Application/Task:** To the best of our knowledge, our work is the first to tackle the novel task of **vocabulary-free FGVR with only a few unlabeled samples as natural observation**. To tackle this novel task we harness the reasoning capabilities of LLMs and leverage the Internet-scale expert knowledge encoded in their weights to first discover the fine-grained categories that are present in a dataset, which is otherwise not know a priori. While CaSED (Conti et al., 2023) previously addressed vocabulary-free classification by retrieving concepts from a knowledge-base, our method is tailored specifically for FGVR by leveraging the knowledge in LLM. This also resonates with the comments from `Reviewer 3hkC` and `Reviewer WYuk` who commented that utilization of LLM for FGVR as "**a new way**" and "**a promising way**", respectively.
> >
> >* **Methodological Advancements:** We introduced **FineR**, a novel system that integrates the visual information translation capability of VQA models, the reasoning capability of LLMs, and the zero-shot discrimination capability of contrastive VLMs to address the vocabulary-free FGVR task. This approach is recognized by `Reviewer WYuk` as "**novel**" and "**interesting**". Furthermore, as shown in the preliminary experiments in Fig. 2, we observed that other LLM-based methods requiring training alignments (such as BLIP-2 learning Q-former, LLaVA, and MiniGPT-4 learning a projector) struggle to discover fine-grained categories effectively due to the limitations of alignment data. These training-based methods cannot fully utilize the reasoning capabilities of LLMs. In contrast, our FineR system translates visual information from unlabeled images into natural language in order to fully harness the reasoning power of LLMs for fine-grained concepts. Thus, we showed that visual and textual concepts can be aligned *without* the need of expensive training.
> >
> >* **Resource Efficiency and Accessibility:** Our approach is both training-free and vocabulary-free, significantly reducing the computational and data resources typically required for vision model and LLM alignments. This enhances the accessibility and feasibility of our method for a broader range of users, potentially democratizing the application of FGVR across various sub-fields.
> >
> >* **Scalability and Generalization:** Being training-free enhances the scalability and generalizability of our methods. As demonstrated in Fig. 7 (Section 3.2, page 8) and Fig. 24 (Appendix K.3, page 31), our method can effectively discover wild and novel fine-grained Pokémon categories, outperforming other methods that fail in these wild scenario.
> > > We have summarized our novelty and contributions at the end of the Introduction section (Sec. 1) in the revision.
>
> Thank you once again for your insightful and helpful advice. We have integrated your suggestions into the revised manuscript, which we believe has led to an improved submission. With the specific changes made in response to your suggestions, such as "Adding citations of some key related works" and "Providing more examples or failure cases for Fig. 2", we hope these improvements in the revision will encourage you to raise your confidence.

---

> > ### Comment · Reviewer_oRxX · 2023-11-18
> > **Good paper**
> >
> > I have checked the reviews of all my felllow reviewers and the corresponding rebuttals. The concerns seem to be solved in a satisfactory way. My own concern is also well-addressed. By the way, the last question “What is your view of NOVELTY of a work using LLMs without training anything?” is not doubt your novelty, instead, I just want to discuss with you about this aspect, and I appreciate your novelty.
> >
> > In conclusion, I agree with the WYuk’s acceptance recommendation, insist my 8 score, and raise the confidence to 5.

---

> > > ### Author Response · Authors · 2023-11-19
> > > **Thanks for your acknowledgement**
> > >
> > > Dear Reviewer oRxX,
> > >
> > > We greatly appreciate your insightful comments, as well as for your satisfaction with our responses and your support of our paper. We are also pleased that our response has increased your confidence. We shall incorporate the aforementioned important discussions into the final manuscript. We really enjoy communicating with you and appreciate your efforts.

---

### Author Response · Authors · 2023-11-18
**Global Response to All Reviewers**

We sincerely thank the ACs and reviewers for their dedicated efforts in reviewing our paper.

We thank all reviewers for their positive, thoughtful and helpful feedback, which we believe has led to a substantially stronger paper with the revision now posted.

We are encouraged that they found our work to be "Novel" (```Reviewer oRxX``` and ```Reviewer WYuk```), "Interesting" (```Reviewer WYuk```), "a new way to do training-free FGVC based on LLM" (```Reviewer WYuk```) and "a promising way ..." (```Reviewer 3hkC```), and our method as "Good performance" (```Reviewer oRxX``` ) and "experimentally better than existing state-of-the-art methods" (```Reviewer WYuk```). We are glad ```Reviewer oRxX```  recognized our work as "The utilization of LLM to vocabulary-free FGVR tasks is novel" and "The well utilization of LLM". Furthermore, we are sincerely encourgaged  that ```Reviewer WYuk``` recoginized our work as "the novelty and technique score is satisfactory for ICLR". We appreciate all the suggestions and feedbacks.


We have thoroughly addressed all the concerns raised by the reviewers in our responses. Additionally, we have integrated the suggestions made by `Reviewers oRxX, 3hkC, and WYuk` into the revised manuscript, with these changes highlighted in red for clarity. The summary of the modifications is as follows:


- We have better clarified our novelty and contribution at the end of the Introduction section (Sec. 1).

-  We have included an in-depth discussion in Appendix K.4. to compare our setting of **vocabulary-free FGVC with only a few unlabeled samples** to Generalized Zero-Shot Transfer (`Reviewer 3hkC`), and VLM Prompt Tuning and Enrichment (`Reviewer WYuk`, `Reviewer oRxX`).

- We have included the discussion on two related work suggested by `Reviewer oRxX` in the Related Work section (Sec. 4).

- We have added a further qualitative failure case analysis in Appendix I, following  `Reviewer oRxX`'s suggestion.

- We have added a more detailed flow diagram (Figure.11 in Appendix B.2) to clearly illustrate our enture recognition pipeline with actual inputs and outputs, following  `Reviewer 3hkC`'s suggestion.

We expect ACs and reviewers to fully consider the following factors when making the final decision: (1) We proposed a novel vocabulary-free FGVR task; (2) A novel utilization of LLM reasoning capability to tackle this novel task, as recognized by both ```Reviewer oRxX``` and ```Reviewer WYuk```; (3) Good performance (`Reviewers oRxX and WYuk`); (4) Comprehensive responses to all the reviewers’ comments, and (5) We admit to release the source code.

We thank all the ACs and reviewers for their time and effort! Please let us know if you have any additional questions or concerns. We are happy to provide clarification.

Authors of Submission #2720

---

### Meta-Review · Area_Chair_A6Zk · 2023-12-06

**Metareview:**

All reviewers lean positive; the rebuttal addressed most concerns.

**Justification For Why Not Higher Score:**

only one 8

**Justification For Why Not Lower Score:**

no score below 6

---

### Decision · Program_Chairs · 2024-01-16

Accept (poster)